# Structure of the chromatin remodelling enzyme Chd1 bound to a ubiquitinylated nucleosome

Ramasubramanian Sundaramoorthy[1], Amanda L Hughes[1], Hassane El-Mkami[2], David G Norman[3], Helder Ferreira[4], Tom Owen-Hughes[1]*

[1]Centre for Gene Regulation and Expression, School of Life Sciences, University of Dundee, Dundee, United Kingdom; [2]School of Physics and Astronomy, University of St Andrews, St Andrews, United Kingdom; [3]Nucleic Acids Structure Research Group, University of Dundee, Dundee, United Kingdom; [4]School of Biology, University of St Andrews, St Andrews, United Kingdom

**Abstract** ATP-dependent chromatin remodelling proteins represent a diverse family of proteins that share ATPase domains that are adapted to regulate protein–DNA interactions. Here, we present structures of the *Saccharomyces cerevisiae* Chd1 protein engaged with nucleosomes in the presence of the transition state mimic ADP-beryllium fluoride. The path of DNA strands through the ATPase domains indicates the presence of contacts conserved with single strand translocases and additional contacts with both strands that are unique to Snf2 related proteins. The structure provides connectivity between rearrangement of ATPase lobes to a closed, nucleotide bound state and the sensing of linker DNA. Two turns of linker DNA are prised off the surface of the histone octamer as a result of Chd1 binding, and both the histone H3 tail and ubiquitin conjugated to lysine 120 are re-orientated towards the unravelled DNA. This indicates how changes to nucleosome structure can alter the way in which histone epitopes are presented.

DOI: https://doi.org/10.7554/eLife.35720.001

*For correspondence:
t.a.owenhughes@dundee.ac.uk

Competing interests: The authors declare that no competing interests exist.

## Introduction

The extended family of ATPases related to the yeast Snf2 protein acts to alter DNA-protein interactions (*Flaus et al., 2006*; *Narlikar et al., 2013*). They act on a diverse range of substrates. For example, while the Mot1 protein acts on complexes between the TATA box binding protein BP and DNA (*Wollmann et al., 2011*), the Snf2 protein carries out ATP-dependent nucleosome disruption (*CoteCôté et al., 1994*). At the heart of all these proteins are paired domains capable of rearranging during the ATP hydrolysis cycle to create a ratchet like motion along DNA in single base increments (*Clapier et al., 2017*; *Gu and Rice, 2010*; *Velankar et al., 1999*).

The yeast Chd1 protein is a member of this protein family and acts to organise nucleosomes over coding regions (*Gkikopoulos et al., 2011*; *Ocampo et al., 2016*; *Pointner et al., 2012*; *Tran et al., 2000*). Consistent with this, Chd1 is known to interact with elongation factors including the Spt4-Spt5 proteins, Paf1 and FACT (*Kelley et al., 1999*; *Krogan et al., 2002*; *Simic et al., 2003*). The partially redundant functions of Chd1 and Isw1 in organising nucleosomes over coding regions are in turn required to prevent histone exchange and non-coding transcription (*Hennig et al., 2012*; *Radman-Livaja et al., 2012*; *Smolle et al., 2012*).

In addition to the positioning of nucleosomes, the distribution of many histone modifications is ordered with respect to promoters (*Liu et al., 2005*; *Mayer et al., 2010*). For example, histone H3 K4 methylation is frequently observed at promoters, while histone H3 K79 and K36 trimethylation are detected in coding regions (*Kizer et al., 2005*; *Li et al., 2003*; *Pokholok et al., 2005*). Histone

**eLife digest** The DNA inside cells contains all the information needed to build an organism. Human DNA measures about 2 metres. To condense it, DNA is wrapped around eight histone proteins to form disc-like structures, called nucleosomes. Nucleosomes are further compressed into chromatin fibres that make up our chromosomes.

The way DNA is packaged and positioned into the nucleosomes can be variably controlled and affects how genes are switched on and off. Although all cells have the same DNA, the way specific genes are turned on and off gives rise to the different types of cells in our body. Specialised motor proteins, called 'chromatin remodellers', control the positioning of nucleosomes inside cells. In yeast cells, for example, the protein Chd1 moves nucleosomes along the DNA so that they are evenly spaced. So far, it has been unclear how chromatin remodellers interact with nucleosomes.

To investigate this further, Sundaramoorthy et al. studied the structure of Chd1 bound to a nucleosome, in which one histone protein was modified with a molecule, called ubiquitin, which is present on genes where Chd1 is known to be active. The structure revealed that both Chd1 and the nucleosome did not have their usual shape. Moreover, Chd1 partially unwrapped DNA from the nucleosomes. As a consequence, the ubiquitin moved to interact with the unwrapped DNA; as did a flexible area on one of the histones, known as 'histone tail'. Both ubiquitin and histone tails play important roles in signalling processes on chromatin. Therefore, such a rearrangement could affect the transmission of signals from chromatin.

The organisation of nucleosomes affects the accessibility of the underlying DNA. As a result, any process that happens on DNA is affected – including controlling when genes are turned on and off under normal conditions, and when things go wrong during diseases. A better knowledge of how the organisation of nucleosomes is controlled will improve our understanding of gene regulation.
DOI: https://doi.org/10.7554/eLife.35720.002

H2B is also observed to be ubiquitinylated within coding regions (*Fleming et al., 2008*; *Xiao et al., 2005*). Ubiquitinylation of histone H2B at lysine 123 in budding yeast, H2B K120 (H2BK120ub) in mammals, is dependent on the E2 ligase Rad6 (*Robzyk et al., 2000*) and the E3 ligase Bre1 (*Hwang et al., 2003*; *Wood et al., 2003*) and removed by the deubiquitinases Ubp8 and Ubp10 (*Bonnet et al., 2014*; *Schulze et al., 2011*; *Wyce et al., 2007*). A specific reader of H2BK120ub has not been identified. However, H2BK120ub does assist the histone chaperone FACT in enabling transcription through chromatin (*Pavri et al., 2006*), and has been found to be required for methylation of histone H3 K4 and K79 (*Sun and Allis, 2002*). An intriguing aspect of H2BK120ub is that while mutation of the writer enzymes or K120 itself disrupts nucleosome organisation, deletion of the deubiquitinylases increases chromatin organisation (*Batta et al., 2011*). One way in which H2BK120ub may influence nucleosome organisation is via effects on enzymes responsible for chromatin organisation. Consistent with this H2BK120ub increases nucleosome repositioning mediated by Chd1 (*Levendosky et al., 2016*).

Yeast Chd1 serves as a useful paradigm in that it functions predominantly as a single polypeptide. In addition, the catalytic core of the enzyme has been crystallised in association with the adjacent tandem chromodomains (*Hauk et al., 2010*). Similarly the C-terminal region of the protein has been crystalized revealing that this region includes SANT and SLIDE domains that comprise the DNA binding domain (DNABD) (*Ryan et al., 2011*; *Sharma et al., 2011*) and are also present in ISWI proteins (*Grüne et al., 2003*). Chd1 enzyme engages nucleosomes in a conformation in which the SANT and SLIDE domains bind linker DNA, while the ATPase domains engage DNA at super helical location (SHL) 2 (*Nodelman et al., 2017*; *Sundaramoorthy et al., 2017*). Higher resolution structures of Chd1 (*Farnung et al., 2017*), Snf2 (*Liu et al., 2017*) and INO80 (*Ayala et al., 2018*; *Eustermann et al., 2018*) show that the ATPase domains make contacts with DNA via residues that are conserved in ancestral single-stranded ATPases and some unique to Snf2-related ATPases. The binding of the Chd1 DNABD unravels two turns of DNA from the surface of nucleosomes in a nucleotide-stimulated reaction (*Farnung et al., 2017*; *Sundaramoorthy et al., 2017*). Here, we report a structure for the yeast Chd1 protein in association with a nucleosome, bearing modifications that are found to occur within coding regions, where Chd1 is known to act. Interestingly, nucleosomal

epitopes are observed to be reconfigured specifically on the side of the nucleosome on which DNA is unwrapped. This indicates the potential for changes to nucleosome structure to reconfigure the way in which histone epitopes are presented.

## Results

### The structure of Chd1 nucleosome complexes

As Chd1 functions on transcribed genes, it is of interest to understand the interplay between Chd1 and histone modifications observed in coding region chromatin. As a result, nucleosomes were prepared in which histone H3 K36 was alkylated to mimic trimethylation (*Figure 1—figure supplement 1*) and H2B cross-linked to ubiquitin (*Figure 1—figure supplement 2*). Conditions were established to favour binding of a single Chd1 to modified nucleosomes that included an asymmetric linker DNA extension of 14 bp (*Figure 1—figure supplement 2B*) in the presence of ADP-BeF. Purified complexes were frozen onto EM grids.

2D classification of some 893000 particles revealed 16 classes in which nucleosomes with the Chd1 molecule attached could be identified (*Figure 1—figure supplement 3B*). Initial 3D classification resulted in five related classes (*Figure 1—figure supplement 3C*). Three of these were combined and reclassified as six classes, one of which was selected for refinement. This resulted in the generation of a map with an average resolution of 4.5 Å (FSC 0.143) (*Figure 1—figure supplement 4A*). The resolution varies within the map, with resolution highest in the region occupied by the nucleosome and ATPase lobes and lower resolution in the vicinity of the DNABD and ubiquitin peptides (*Figure 1—figure supplement 4B*). The nucleosome particles exhibited a preferred orientation, which may limit the resolution (*Figure 1—figure supplement 4C*). A structural model was generated to fit the density map making use of the structures of a nucleosome assembled on the 601 DNA sequence, Chd1 chromoATPase, and DNABD (*Figure 1*). The fit for individual components of the structure to the electron density is shown in *Figure 1—figure supplement 5*.

The overall organisation of Chd1 is similar to that observed previously by cryo EM (*Farnung et al., 2017*; *Sundaramoorthy et al., 2017*) and directed cross-linking (*Nodelman et al., 2017*). The ATPase domains are bound at the SHL-2 location. Of the two SHL2 locations within nucleosomes, the bound site is in closest proximity to SANT-SLIDE domain bound linker DNA in physical space, but distal on the unwrapped linear DNA sequence (*Figure 1*). Chd1 predominantly contacts the nucleosome via contacts with DNA, via the DNABD in the linker and ATPase lobes at SHL2; contacts with histones are limited to the histone H3 and H4 N-terminal regions discussed below.

We previously showed that Chd1 binding results in nucleotide-dependent unwrapping of nucleosomal DNA resulting from the interaction of the DNABD with linker DNA (*Sundaramoorthy et al., 2017*). The higher resolution of the current structure shows that precisely two turns of nucleosomal DNA are unravelled (*Figure 1*). The extent of DNA unwrapping observed here when Chd1 is bound to nucleosomes flanked by a 14 base pair linker DNA is identical to that observed when Chd1 is bound to the opposite surface of the 601 nucleosome positioning sequence with a 63 base pair linker (*Farnung et al., 2017*). As the interaction of histones with the two sides of the 601 positioning sequence differ quite dramatically (*Chua et al., 2012*; *Hall et al., 2009*; *Levendosky et al., 2016*; *Ngo et al., 2015*), this suggests that the extent of unwrapping is dominated by the properties of Chd1 rather than the affinity of DNA for the octamer. The path of this unwrapped DNA is oriented away from the plane of the wrapped DNA gyre and is kinked at the location where contacts are made with the SANT-SLIDE domains (*Figure 1*). Other than DNA unwrapping, we do not detect additional changes in the organisation of DNA on Chd1 bound nucleosomes at this resolution.

The orientation of the DNABD is critical in determining the extent of DNA unwrapping. The only contacts detected between the DNABD and the remainder of Chd1 are contacts with the chromodomains (*Figure 2* contact I and II). The first of these is the interaction between K329 of chromodomain II and D1201 P1202 in the SLIDE component of the DNABD and has been observed previously (*Farnung et al., 2017*; *Nodelman et al., 2017*) (*Figure 2—figure supplement 1A*). The second contact is between S344 and K345 in the linker helix between chromodomain II and ATPase lobe I with the SANT component of the DNA binding domain at D1033-D1038 (*Figure 2—figure supplement 1A*). Given that chromodomains are present in Chd1 enzymes but not ISWI and Snf2 remodellers, it

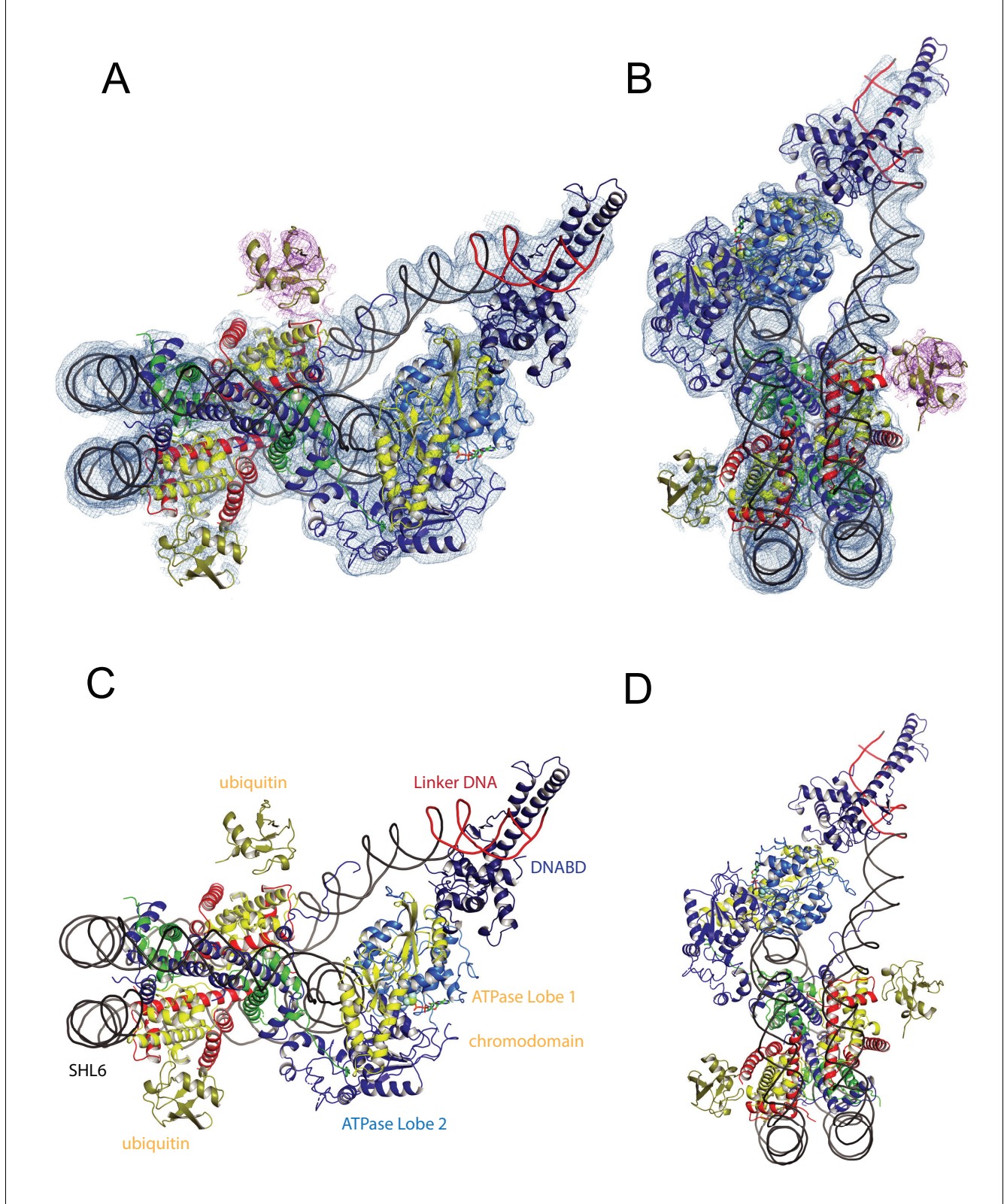

**Figure 1.** A Chd1-Nucleosome complex. (**A, B**) Overall fit of nucleosome bound Chd1 to density map. Chd1 chromodomains – yellow, DNABD – dark blue, ATPase lobe one cyan, ATPase lobe two blue, Ubiquitin dark yellow, H2B yellow, H2A red, H3 green, H4 blue. (**C, D**) Two views of the structural model.

DOI: https://doi.org/10.7554/eLife.35720.003

*Figure 1 continued on next page*

*Figure 1 continued*

The following source data and figure supplements are available for figure 1:

**Source data 1.** Raw data for *Figure 1—figure supplement 1C*.
DOI: https://doi.org/10.7554/eLife.35720.009
**Figure supplement 1.** Preparation of nucleosomes with an alkylation mimic of histone H3 K36 methylation.
DOI: https://doi.org/10.7554/eLife.35720.004
**Figure supplement 2.** Generation of nucleosomes with ubiquitin cross-linked to histone H2B K120.
DOI: https://doi.org/10.7554/eLife.35720.005
**Figure supplement 3.** Image processing for Chd1-nucleosome complex.
DOI: https://doi.org/10.7554/eLife.35720.006
**Figure supplement 4.** Properties of the Chd1-nuclesome complex.
DOI: https://doi.org/10.7554/eLife.35720.007
**Figure supplement 5.** Fit of individual components.
DOI: https://doi.org/10.7554/eLife.35720.008

makes sense that the residues contacted in the SANT and SLIDE domains are most highly conserved in Chd1 proteins (*Figure 2—figure supplement 1B*) (*Hall et al., 2009*; *Meng et al., 2015*; *Sundaramoorthy et al., 2017*).

## Repositioning of Chd1 ATPase lobes to a closed ATP-bound state drives repositioning of chromodomains

The position of the chromodomains is determined by each of the four contacts made with other components of the complex (*Figure 2*). When not bound to nucleosomes, the tandem chromodomains of Chd1 are observed to impede DNA binding to the ATPase domains (*Hauk et al., 2010*). This gave rise to the prediction that these domains would be rearranged in the nucleosome-bound state (*Hauk et al., 2010*). This is indeed the case as the chromodomains undergo an 18 degree rotation when compared to the orientation observed in the crystal structure of Chd1 in the open state (*Figure 3*). Following repositioning, chromodomain I interacts with nucleosomal DNA at SHL1 (*Figure 2—figure supplement 2*) as observed previously (*Farnung et al., 2017*; *Nodelman et al., 2017*).

Coincident with repositioning of the chromodomains, ATPase lobe II is repositioned closer to lobe I. This results in residues including those contributing to the conserved Walker box motifs (K407 and R804, R807) being brought into an arrangement compatible with ATP catalysis. Density for ADP-BeF within the pocket formed by conserved residues from ATPase domains I and II is well defined (*Figure 2—figure supplement 3*).

The repositioning of ATPase lobe II enables contacts to be made with nucleosomal DNA (see below), the histone H4 tail and the histone H3 alpha one helix (*Figure 2—figure supplement 4*). These are the only direct contacts with histone components of the nucleosome. The contact with the H4 tails is conserved in mtISWI and Snf2 (*Liu et al., 2017*; *Yan et al., 2016*). D729 and E669 are conserved across all classes of remodelling enzyme but D725 is not as well conserved in Snf2-related enzymes (*Figure 2—figure supplement 4B*). The conservation of this contact in Chd1 enzymes is consistent with the H4 tail playing an important role in regulating Chd1 activity; deletion or mutation of the H4 tail has been shown to reduce nucleosome sliding and ATPase activity (*Ferreira et al., 2007*).

The additional helices that make up the protrusion 2 region of ATPase lobe two in Chd1 are conserved in chromatin remodeling ATPases, but not within all SF2 DNA translocases. In the crystal structure of the Chd1 chromoATPase and the previously published Chd1-nucleosome structure this region was not mapped (*Farnung et al., 2017*; *Hauk et al., 2010*). However, within the structure presented here residues ranging from 632 to 647 pack against the alpha 1 helix of histone H3. In particular the two conserved residues K632 and K642 are located closer to H3 $\alpha$1 D81 and E73 (*Figure 2—figure supplement 4A*). Deletion of this lobe two loop K632-K646 abolishes nucleosome sliding consistent with a role for this region in Chd1 action (*Figure 2—figure supplement 5*). The residues participating in the interaction are progressively less well conserved in ISWI and SNF2 related proteins (*Figure 2—figure supplement 4C*). A loop from Phe1033 to Leu 1045 in an

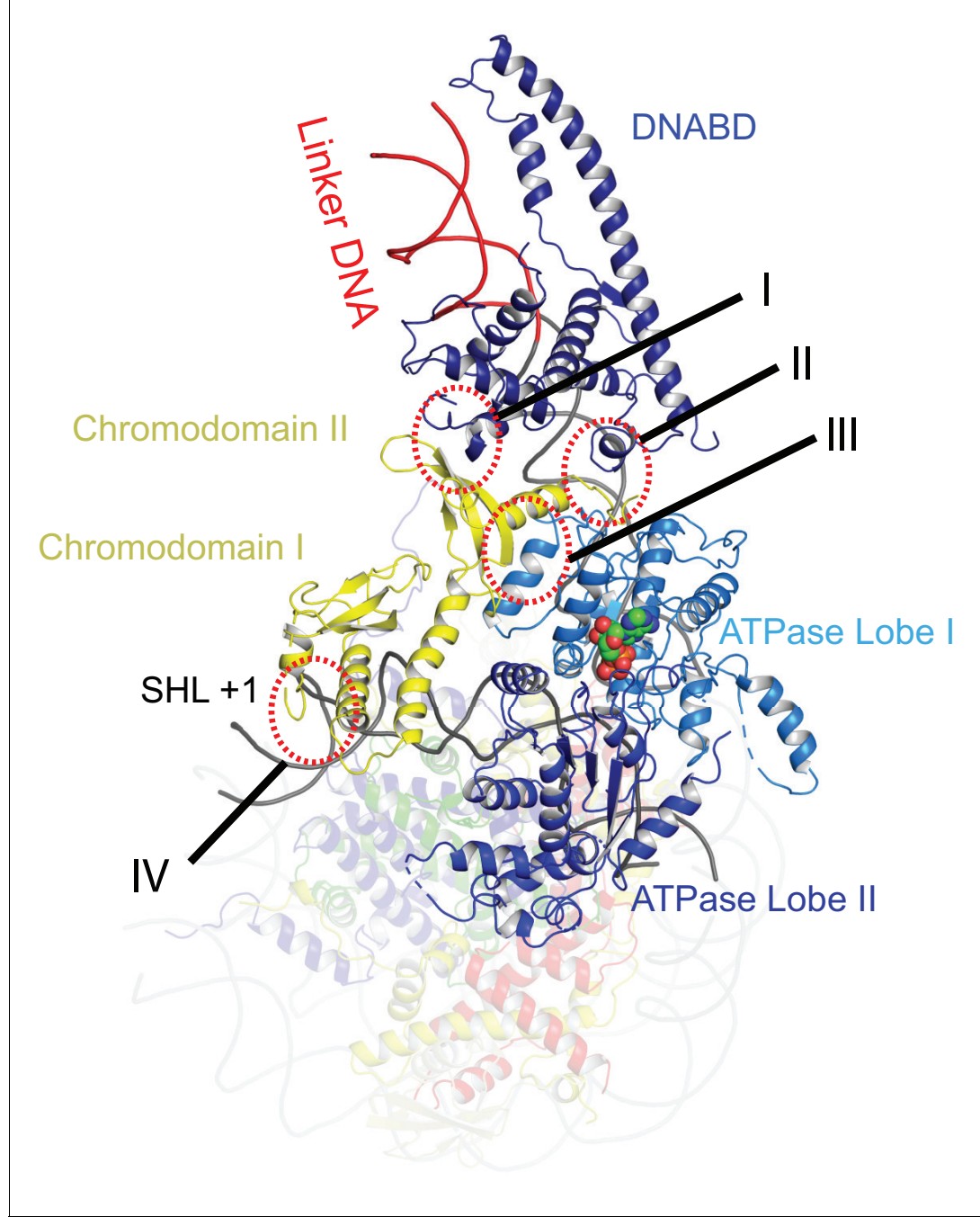

**Figure 2.** Contacts constraining Chd1 chromodomains. Overview of the major contacts constraining the positioning of the Chd1 chromodomains. Colours of domains as for *Figure 1*. Key contacts are highlighted. (I) Chromodomain II SLIDE, (II) Chromodomain linker helix to SANT, (III) Chromodomain II to ATPase lobe 1, (IV) Chromodomain I to nucleosomal DNA at SHL + 1.

DOI: https://doi.org/10.7554/eLife.35720.010

The following source data and figure supplements are available for figure 2:

**Source data 1.** Raw data for *Figure 2—figure supplement 5A*.
DOI: https://doi.org/10.7554/eLife.35720.016

**Figure supplement 1.** The chromodomain contacts with SANT and SLIDE domains.
DOI: https://doi.org/10.7554/eLife.35720.011

**Figure supplement 2.** Chromodomain interactions with nucleosomal DNA at SHL + 1.
DOI: https://doi.org/10.7554/eLife.35720.012

**Figure supplement 3.** The ATP binding site.

*Figure 2 continued on next page*

*Figure 2 continued*

DOI: https://doi.org/10.7554/eLife.35720.013

**Figure supplement 4.** ATPase lobe two interaction with histone H4 tail.

DOI: https://doi.org/10.7554/eLife.35720.014

**Figure supplement 5.** Mutation of ATPase residues contacting histone H3 abolishes Chd1 activity.

DOI: https://doi.org/10.7554/eLife.35720.015

equivalent region of the yeast Snf2 protein is not assigned in the Snf2-nucleosome structure, but this region is positioned such that a related contact with histone H3 could be made.

The structure, also provides clues as to how these conformational changes are driven. A central event is likely to be the closure of the cleft between ATPase domains driven by ATP binding (*Figure 2—figure supplement 3*). The 40° rotation of ATPase lobe II required to form the ATP binding pocket results in a positively charged surface, observed to interact with an acidic surface on the long helix of chromodomain I (*Figure 3A*) (*Hauk et al., 2010*), being replaced by an acidic surface likely to repel chromodomain I (*Figure 3B*). As a result, closure of the ATPase domains is anticipated to drive nucleotide-dependent repositioning of the chromodomains. Pulsed EPR was used to directly measure repositioning of the chromodomains in the absence of nucleosomes (*Figure 4*). The distance between engineered labels at V256C in chromodomain I and S524C in ATPase lobe1 is 4.4 nm in the open state, consistent with that observed in the crystal structure of the Chd1 chromoATPase domains (*Hauk et al., 2010*). In the presence of ADP-BeF the 4.4 nm distance predominates, but a shoulder is observed consistent with a proportion of molecules adopting a new conformation with a distance of 5.6 nm (*Figure 4*) which is similar to that observed in the ADP-BeF bound nucleosome by cryo-EM. This indicates that ATP binding is a driving event for repositioning of the chromodomains.

The partial repositioning of the chromodomains observed in free Chd1 is likely to be stabilised by additional favourable interactions formed when this repositioning occurs within the context of nucleosome bound Chd1. These include the formation of contacts between chromodomain I and DNA at SHL1, between ATPase lobe II and the H3 alpha one helix, between ATPase lobe II and the histone H4 tail and most significantly the formation of a substantial interaction interface between ATPase lobe II and nucleosomal DNA at SHL2. The repositioning of the chromodomains in turn acts as a lever to reposition the DNA binding domain. In the context of nucleosomes this results in nucleotide-dependent unwrapping of two turns of nucleosomal DNA (*Sundaramoorthy et al., 2017*). Conversely, the interaction of the DNABD requires linker DNA to be accessible.

In order to investigate how the ability of the DNABD to interact with linker DNA is affected by the presence of an adjacent nucleosome, interactions between dinucleosomes with different separations were modelled. With a linker length of 19 bp Chd1 can be modelled binding the linker between adjacent nucleosomes (*Figure 5*). However, as the linker between nucleosomes is reduced, steric clashes become increasingly prohibitive. The requirement for a 19 bp linker is likely to provide a limit below which engagement of the DNABD will be less stable. As this lower limit is set by clashes between the DNABD and the adjacent nucleosome, it is different from the length of linker required to occupy the DNA-binding surface of the SANT and SLIDE domains on a mononucleosome with a free DNA linker. In this latter case, seven base pairs of DNA make contact with the DNABD (*Figure 1*). The c19 bp separation below which access of the DNABD to linker becomes progressively more difficult resonates with the average inter-nucleosome spacing of 19 bp observed in *Saccharomyces cerevisiae* (*Tsankov et al., 2010*). As the conformation of the DNABD is connected via the chromodomains to the ATPase domains, the structure of Chd1 provides molecular connectivity between the availability of nucleosomal linker DNA in excess of 19 bp and the generation of closed nucleotide bound motor domains. This potentially provides a mechanism via which linker DNA length regulates the rate of nucleosome movement (*Yang et al., 2006*).

## Organisation of the Chd1 ATPase domains

Nucleosome repositioning is likely to be driven by the ability of the ATPase domains to drive ATP-dependent DNA translocation. This has been observed directly for several Snf2 family proteins (*Deindl et al., 2013*; *Lia et al., 2006*; *Sirinakis et al., 2011*; *Zhang et al., 2006*) and is conserved within a wider family of superfamily II ATPases (*Singleton et al., 2007*). Structures of superfamily II

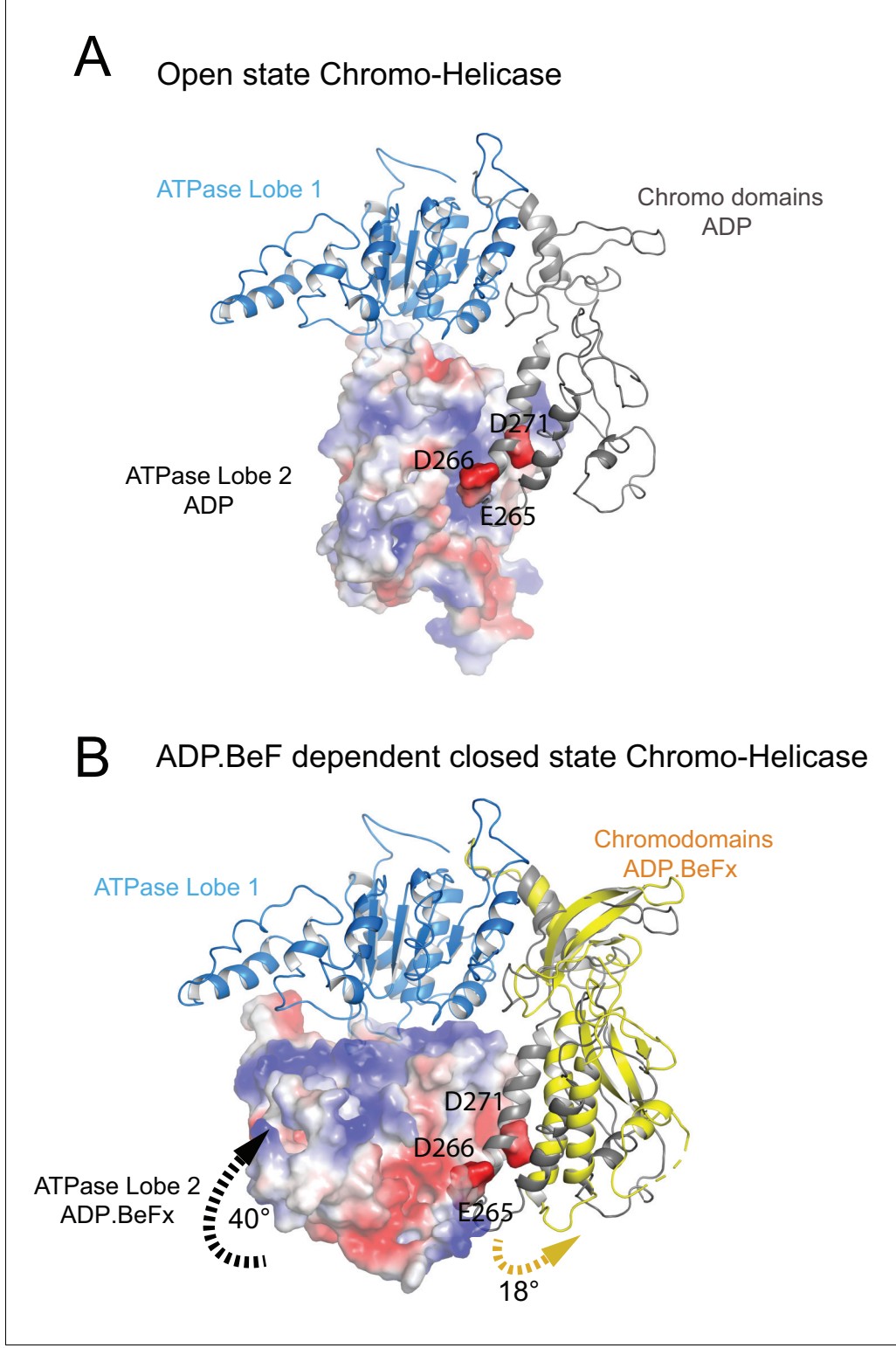

**Figure 3.** Closure of the ATPase lobes changes the chromodomain interaction surface. (**A**) The long acidic helix within chromodomain I interacts with a basic surface on ATPase lobe two in the open state (3MWY). (**B**) In the closed state, the basic surface on lobe two is rotated towards DNA and replaced with an acidic region. The long acidic helix within chromodomain I is repositioned away from this acidic surface.

DOI: https://doi.org/10.7554/eLife.35720.017

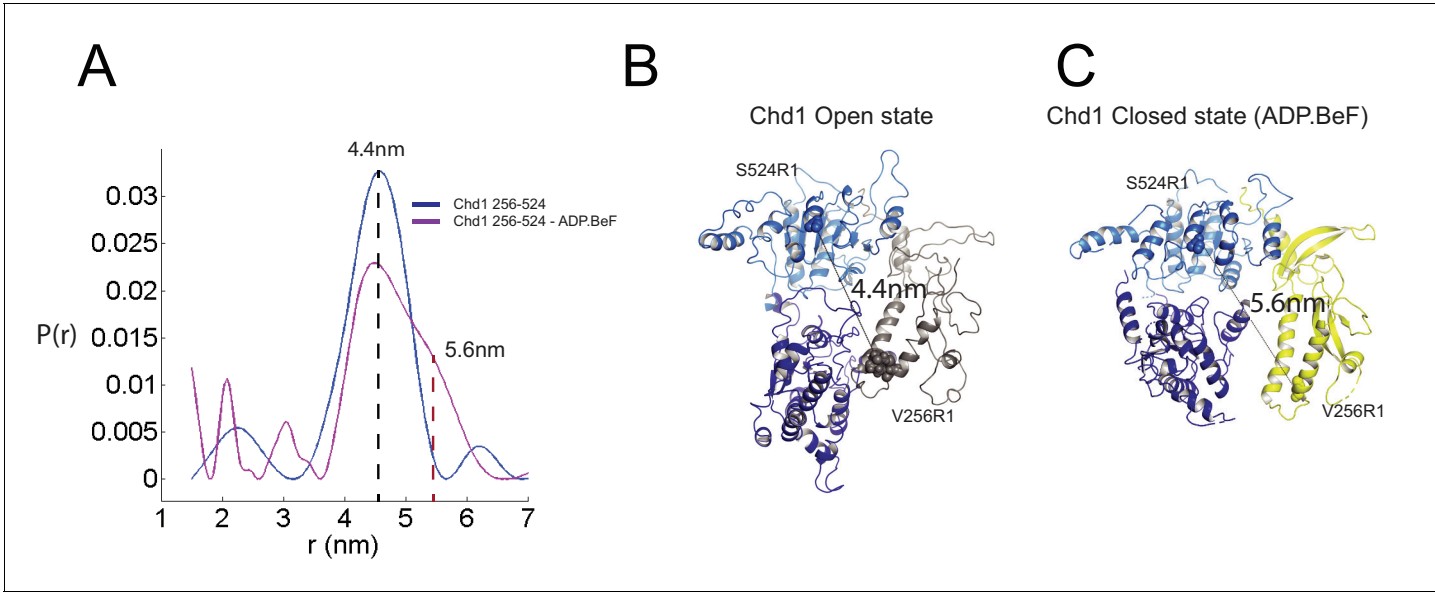

**Figure 4.** Nucleotide dependent reconfiguration of Chd1 chromodomains. Pulsed electron paramagnetic measurements were used to measure the distance between nitroxyl reporter groups attached to Chd1 at ATPase lobe 1 (S524) and chromodomain I (V256). (**A**) The probability distribution P(r) at different separations was measured in the presence (purple) and absence (blue) of ADP.BeF. The distance corresponding to the major distance is shown for both measurements. The distance corresponding to the additional distribution appears as a shoulder in the presence of ADP.BeF$_x$ and is also indicated. Modelled distances between these labelling sites in the open state (3MWY), and the closed state observed in the Chd1 bound nucleosome are indicated in (**B**) and (**C**), respectively.

DOI: https://doi.org/10.7554/eLife.35720.018

The following source data is available for figure 4:

**Source data 1.** PELDORDistanceDistribution.csv.

DOI: https://doi.org/10.7554/eLife.35720.019

single stranded translocases, such as herpes virus NS3, in different NTP bound states illustrate how the ratcheting motion of the ATPase domains drives translocation (*Gu and Rice, 2010*). To date such a series of structures does not exist for a double strand specific translocase. This raises the question as to whether structures of NS3 can be used to inform key aspects of the mechanism of Chd1 such as identifying the tracking strand. To do this we first align the ATPase lobes of Chd1 individually with NS3. The ATPase lobes of Chd1 like other Snf2 related proteins contain additional helices not conserved with NS3 (*Dürr et al., 2005*; *Liu et al., 2017*; *Thomä et al., 2005*). As a result, the alignment is restricted to conserved helices. In the case of lobe I and II, the RMSD of the fit is 4.9 Å and 6.5 Å, respectively (*Figure 6—figure supplement 1A*). In the closed state alignment of both domains with the structure of NS3 in the ADP.BeF bound state results in an RMSD of 9.8 Å. Using this alignment, the ssDNA bound by NS3 can be docked into the Chd1-Nucleosome structure (*Figure 6A*). This ssDNA aligns with the top strand of nucleosomal DNA (*Figure 6*). Conserved motif Ia in ATPase lobe one and motifs IV and V from ATPase lobe two contact this strand. These residues undergo a ratcheting motion during the course of ATP hydrolysis that drives the ssDNA through NS3 (*Gu and Rice, 2010*). Similar motion between these residues would be anticipated to drive nucleosomal DNA across the nucleosome dyad in the direction of the longer linker (*Figure 6B*).

It is notable that within Chd1 additional DNA contacts are made that differ from those observed in NS3. Firstly, motifs II and III within lobe one contact the opposite DNA strand (*Figure 6A*). As these motifs are intimately associated with motif Ia they would be anticipated to undergo a similar ratcheting motion with respect to the contacts made by lobe 2. Secondly, the ATPase lobes of Chd1 like those of Snf2 contain additional helices including the protrusions to the helical lobes and the brace helix that are unique to chromatin remodelling enzymes (*Figure 6A*) (*Farnung et al., 2017*; *Flaus et al., 2006*; *Liu et al., 2017*). These extend the binding cleft between the ATPase lobes and make additional contacts with both DNA strands.

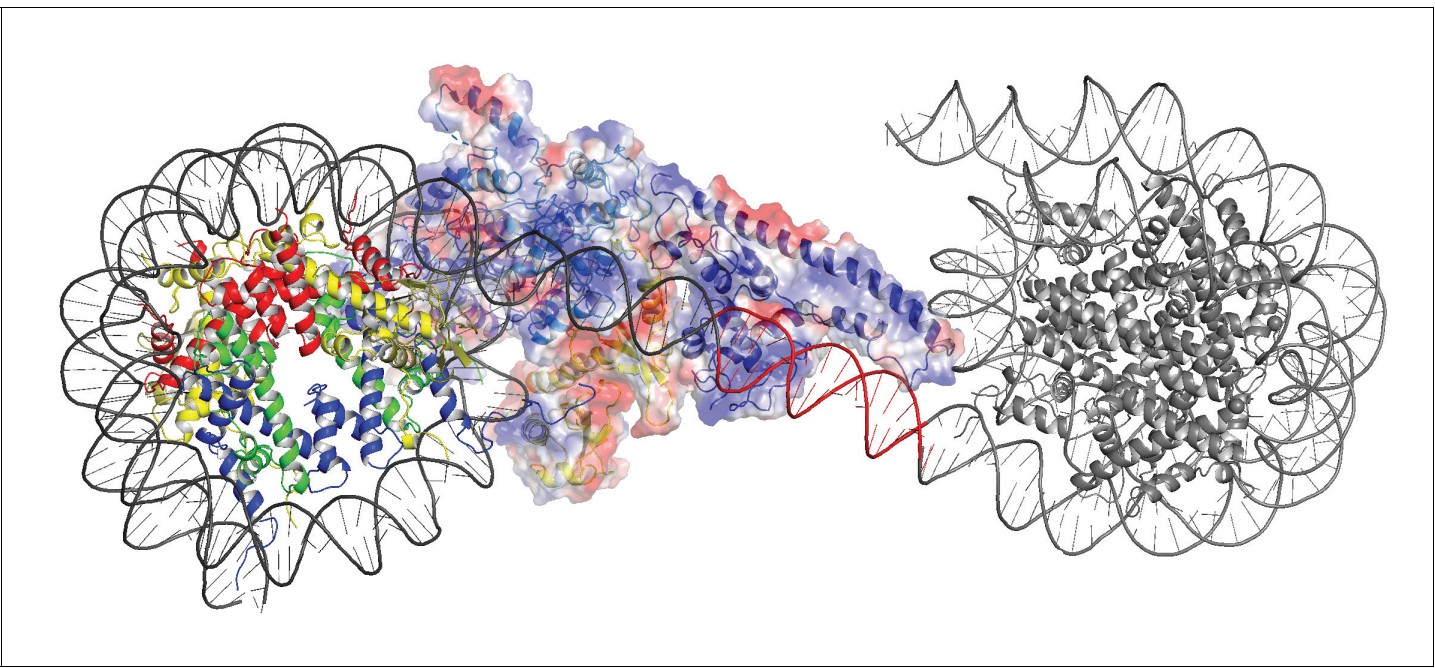

**Figure 5.** Modelling the interaction of Chd1 between adjacent nucleosomes. The structure of the Chd1 bound nucleosome was used to model binding to the linker between an adjacent nucleosome (grey) by extending the liner DNA to 19 bp. With shorter linkers steric clashes with the adjacent nucleosome become progressively more severe.

DOI: https://doi.org/10.7554/eLife.35720.020

The structure of a fragment of the yeast Snf2 protein bound to a nucleosome revealed contacts between ATPase lobe one with DNA at SHL2 and the adjacent DNA gyre at SHL 6 (*Liu et al., 2017*) (*Figure 6—figure supplement 2A*). The basic surface of lobe one responsible for this interaction is not conserved in Chd1, and the acidic residues D464 and E468 make a similar interaction with DNA unlikely. In addition, DNA is not present in this location as it is lifted off the surface of the octamer (*Figure 6—figure supplement 2B*). In the case of the Snf2 protein the interaction with the adjacent DNA gyre is proposed to anchor the translocase preventing it from transiting around the octamer surface (*Liu et al., 2017*). Chd1 has a relatively small interaction interface with histones, so there is a similar requirement for DNA interactions to constrain motion of the whole protein. In the case of Chd1 this could instead be provided through the interaction of the chromodomains with DNA at SHL1 and through the interaction of the DNA binding domain with linker DNA. Amino acids 476 to 480 of lobe one also interact with DNA in the unravelled state (*Figure 6—figure supplement 2B*). These residues are not conserved even in Chd1 proteins so the significance of this contact is not clear.

## Two molecules of Chd1 can bind a single nucleosome using the same mode of binding

Chromatin organising motor proteins are capable of catalysing bidirectional nucleosome repositioning that can occur as a result of the binding of two or one enzyme (*Blosser et al., 2009*; *Qiu et al., 2017*; *Racki et al., 2009*; *Willhoft et al., 2017*). As Chd1 binds to one side of the nucleosome, no steric clashes are anticipated should a second Chd1 bind linker DNA on the opposite side of the nucleosome. To investigate this further, complexes consisting of two Chd1 molecules bound to one nucleosome were prepared using nucleosomal DNA with symmetrical linkers of 14 base pairs and the images processed as indicated (*Figure 7—figure supplement 1*). Most particles were assigned to 2D classes in which two bound Chd1 molecules are discernible, though one is often more dominant likely as a result of the projections of the dominant orientations observed. All 3D classes have two bound Chd1 molecules and the best refined class provides an envelope with 11 Å resolution (FSC 0.143) (*Figure 7*). Two Chd1 molecules bound in the same mode observed in the 1:1 complex

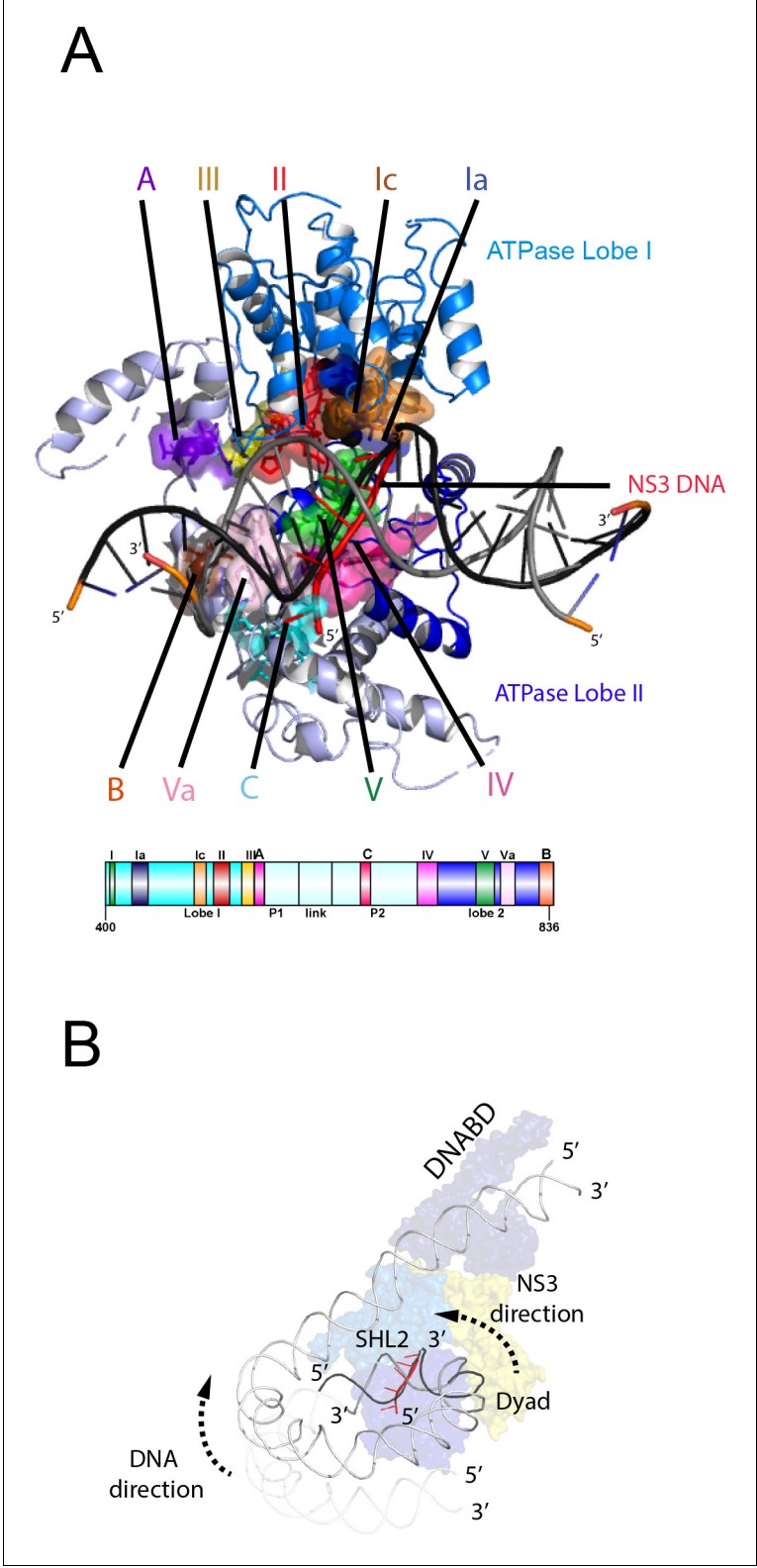

**Figure 6.** Comparison of NS3 and Chd1 interactions with DNA. (**A**) Contacts between motifs conserved with DNA at SHL2. Motifs and colouring are indicated on the structure. The ssDNA from NS3 aligned to Chd1 is shown docked into Chd1 (red). In Chd1 motifs II and III contact the opposite (3′–5′) strand. Contacts made by remodelling enzyme specific extension are labelled A, B and C. The locations of each sequence are indicated in the schematic guide. (**B**) Schematic indicating directionality in context of a nucleosome. The directionality of NS3

*Figure 6 continued on next page*

*Figure 6 continued*

translocation inferred from docking the ssDNA is 3'−5' away from the nucleosome dyad. Assuming movement of Chd1 around the nucleosome is constrained (for example via contact with linker DNA, the H4 tail and histone H3) translocation of Chd1 with this directionality is anticipated to drive DNA in the opposite direction towards the long linker as indicated.

DOI: https://doi.org/10.7554/eLife.35720.021

The following figure supplements are available for figure 6:

**Figure supplement 1.** Comparison of NS3 and Chd1 ATPase domains.
DOI: https://doi.org/10.7554/eLife.35720.022
**Figure supplement 2.** Snf2 and Chd1 interactions with the adjacent DNA gyre.
DOI: https://doi.org/10.7554/eLife.35720.023

can be docked into this volume. There are no direct contacts between the two Chd1 proteins suggesting that the two bound enzymes are likely to function independently. Previously, negative stain EM of two Chd1 molecules bound to a single nucleosome indicated that the DNA binding domain interacted with linker DNA on only one side of the nucleosome (*Nodelman et al., 2017*). Our

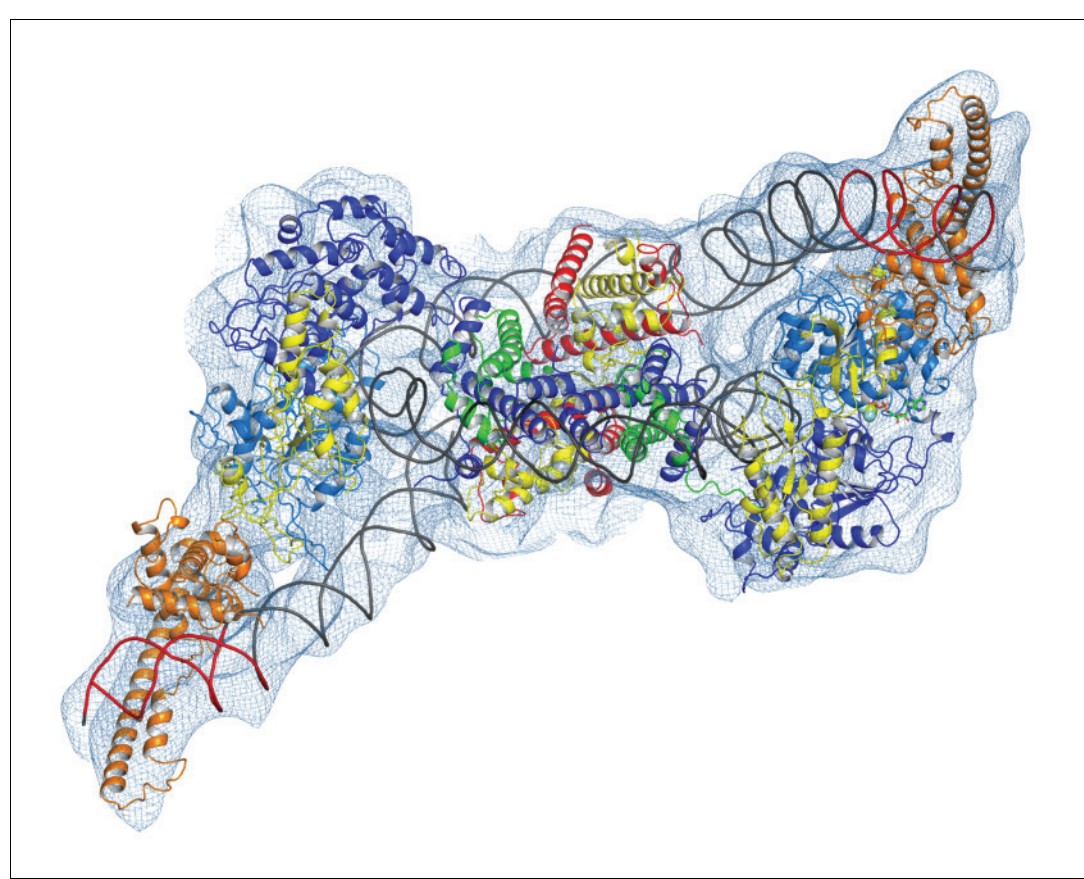

**Figure 7.** Complex of two Chd1 bound to one nucleosome. The volume obtained for the 2Chd1:1Nucleosome complex is shown contoured to 0.0027. Two Chd1 molecules in the conformation observed in *Figure 1* and a nucleosome on which DNA is unwrapped from both sides to the same degree as observed on the Chd1 bound side of the 1:1 complex are shown ridged body fitted into the volume. The correlation coefficient for fitting is 0.95. The colouring of domains is as for *Figure 1*.
DOI: https://doi.org/10.7554/eLife.35720.024
The following figure supplement is available for figure 7:

**Figure supplement 1.** Generation of density map for two Chd1 molecules bound to a single nucleosome.
DOI: https://doi.org/10.7554/eLife.35720.025

envelope shows that both DNA-binding domains can bind to linker DNA simultaneously and that the extent of DNA unwrapping is similar on both sides of the nucleosome. We do however observe that within some 3D classes the path of the unwrapped DNA is less clear on one side of the nucleosome. This could relate to differences in the dynamic interactions of Chd1 with DNA on the two sides of the nucleosome (*Tokuda et al., 2018*).

## The trajectory of the histone H3 tail is altered by DNA unwrapping

On the fully wrapped side of the nucleosome the H3 tail can be traced to proline 38, emerging between the DNA gyres at SHL1. In contrast, on the unwrapped side of the nucleosome the H3 tail can be traced to alanine 26 indicating that on this side of the nucleosome the H3 tail is better ordered. In addition, the trajectory of the tail is different to that observed in previous structures (*Figure 8A*). This altered trajectory was also not observed in a previous structure of a Chd1-bound nucleosome (*Farnung et al., 2017*). This structure was made in the presence of PAF1 and FACT complexes which may have contributed to noise in this region that is not apparent in our structure (*Farnung et al., 2017*). A potential explanation for the defined and altered trajectory of the histone H3 tail on the unwrapped side of the nucleosome is that amino acids within the extreme N-terminal region, that are not resolved in our structure, interact with the unravelled DNA. The 25 N-terminal residues include eight lysine and arginine residues that could interact with DNA at different locations along the unravelled linker. Interestingly, deletion of the H3 tail to H3K36 increases the initial rate at which nucleosomes are repositioned by Chd1 (*Figure 8—figure supplement 1*). This effect is not dependent on DNA binding via the extreme N-terminus as deletion to K26 does not stimulate Chd1 activity (*Figure 8—figure supplement 1*). The region of H3 that exerts the repressive effect occupies the density shown in *Figure 8A*. Surprisingly, no contacts between this region of H3 and Chd1 are observed. It is possible that this region interacts with regions of Chd1 such as the N-terminus that are not resolved, or that the repressive effect is exerted when Chd1 is bound in a conformation different to that observed by EM.

## Ubiquitin interacts with unravelled nucleosomal DNA

The electron density for ubiquitin molecules is not as well defined as other components of the complex, and limiting for the overall resolution (*Figure 1*). This is likely to reflect mobility of the ubiquitin peptides. Consistent with this, the electron density determined from X-ray diffraction on crystallised nucleosomes with ubiquitin conjugated at this location resulted in no density attributable to ubiquitin (*Machida et al., 2016*). On the wrapped side of the nucleosome, ubiquitin is located adjacent to the acidic patch that is widely used as an interface for nucleosome binding proteins (*McGinty and Tan, 2016*) (*Figure 8B*). This is also the location that ubiquitin conjugated to H2A K15 has been observed to occupy on unbound nucleosomes (*Wilson et al., 2016*) and likely represents a favourable conformation for ubiquitin when coupled at different sites within this locality (*Vlaming et al., 2014*).

On the unwrapped side of the nucleosome, the ubiquitin peptide is displaced across the lateral surface towards the DNA. The unwrapped DNA is oriented away from the lateral surface and this positions the DNA backbone at SHL6 in contact with the repositioned ubiquitin (*Figure 8B*). In 2D classes, ubiquitin is more prominent on the unwrapped side suggesting it is more tightly constrained. The residues interacting with DNA include Lys 48 Arg 54 and Asp 60. It is likely this interaction stabilises DNA in the unwrapped state. Using a fluorescence-based assay, H2BK120ub has previously been observed to stimulate the activity of Chd1 118–1274 approximately 2-fold (*Levendosky et al., 2016*). Using a gel-based assay with Chd1 1–1305, we observe a slightly greater 3-fold effect (*Figure 8—figure supplement 2*).

## Discussion

A striking feature of the Chd1 nucleosome complex is the limited number of contacts with histones. The two direct contacts with histones are with the H3 alpha one helix and with the histone H4 tail (*Figure 2—figure supplement 4*). The contact with the H4 tail is required for efficient remodelling by Chd1 (*Ferreira et al., 2007*) as it is for ISWI subfamily enzymes (*Clapier et al., 2001*; *Hamiche et al., 2001*). Aside from these two contacts, the interaction of Chd1 with nucleosomes is dominated by interactions with DNA. This leaves the majority of the nucleosome accessible for

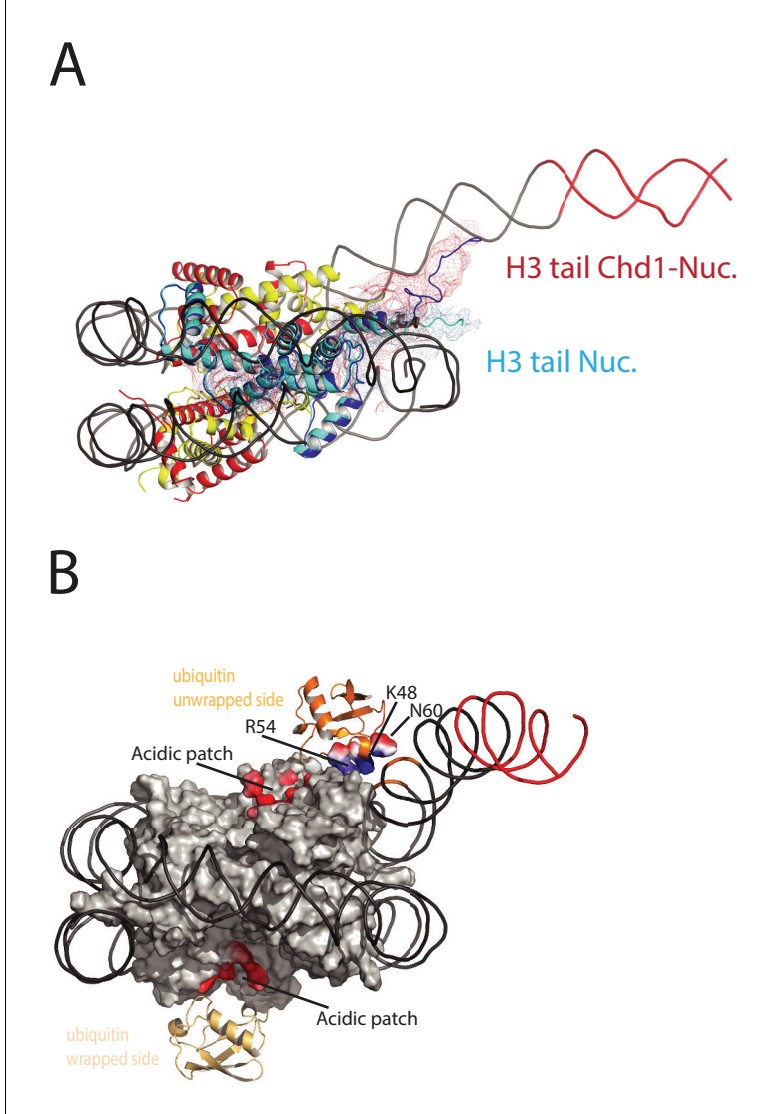

**Figure 8.** Histone epitopes are repositioned on Chd1 bound nucleosomes. (**A**) The histone H3 tail on the unwrapped side of a Chd1 bound nucleosome is shown in dark blue fitted to electron density shown in red. The density extends to residue 26 on this side of the nucleosome and follows a path towards the unwrapped DNA. On the wrapped side of the nucleosome, the H3 tail is less well defined and follows a path similar to that observed on nucleosomes bound by 53BP1 (shown in light blue). H3 tail. (**B**) Ubiquitin on the wrapped side of the nucleosome is located over the acidic patch. On the unwrapped side of the nucleosome it is repositioned away from the acidic patch and interacts with the unravelled DNA.

DOI: https://doi.org/10.7554/eLife.35720.026

The following source data and figure supplements are available for figure 8:

**Source data 1.** Raw data for *Figure 8—figure supplement 1A*.
DOI: https://doi.org/10.7554/eLife.35720.029
**Source data 2.** Raw data for *Figure 8—figure supplement 1B*.
DOI: https://doi.org/10.7554/eLife.35720.030
**Source data 3.** Raw data for *Figure 8—figure supplement 1B(2)*.
DOI: https://doi.org/10.7554/eLife.35720.031
**Source data 4.** Raw data for *Figure 8—figure supplement 1B(3)*.
DOI: https://doi.org/10.7554/eLife.35720.032
**Source data 5.** Raw data for *Figure 8—figure supplement 2*.
DOI: https://doi.org/10.7554/eLife.35720.033
**Figure supplement 1.** The histone H3 tail inhibits Chd1 activity.

*Figure 8 continued on next page*

*Figure 8 continued*

DOI: https://doi.org/10.7554/eLife.35720.027

**Figure supplement 2.** Coupling ubiquitin to Histone H3 K123 directly stimulates Chd1 activity.

DOI: https://doi.org/10.7554/eLife.35720.028

binding by additional factors. We show that a second Chd1 molecule can bind the opposite side of the nucleosome using a similar mode of interaction (*Figure 7*). Even in the case of a nucleosome bound by two Chd1 molecules the lateral surfaces of the nucleosome are accessible for binding by other factors.

Despite a lack of direct contacts with histones H2A and H2B, Chd1 activity is dependent on histone dimers (*Levendosky et al., 2016*). The requirement for histone dimers may arise as a result of dimer loss affecting DNA wrapping. Loss of a histone dimer will result in a loss of histone DNA contacts at SHL3.5, 4.5 and 5.5. More extensive unwrapping of DNA to SHL3.5 would require major repositioning of DNABD in order to retain the interaction with linker DNA while the chromoATPase is engaged at SHL-2. This provides a potential explanation for the dependency of Chd1 activity on histone dimers. Conversely, association of a histone dimer with a histone tetramer or hexamer around which DNA is initially significantly unwrapped, could be stabilised by the binding of Chd1 rewrapping DNA to SHL5. The stabilisation of the SHL5-wrapped state may facilitate correct docking of histone dimers into chromatin. This provides a mechanistic basis for the observed activities of Chd1 in H2A/H2B transfer (*Lusser et al., 2005*) and chromatin assembly (*Fei et al., 2015*; *Lee et al., 2012*; *Torigoe et al., 2013*). Repositioning of the Chd1 DNA-binding domain towards the major orientation observed in free solution, the Apo state reported by (*Sundaramoorthy et al., 2017*), would guide linker DNA towards the fully wrapped state. As a result Chd1 has the potential to function in multiple stages of chromatin assembly and the generation of organised chromatin (*Lusser et al., 2005*; *Robinson and Schultz, 2003*; *Torigoe et al., 2013*).

## Chd1 directionality

The Chd1 enzyme has the ability to organise spaced arrays of nucleosomes both in vitro and in vivo (*Gkikopoulos et al., 2011*; *Lusser et al., 2005*; *Robinson and Schultz, 2003*). Enzymes that exhibit this organising activity typically reposition nucleosomes away from the ends of short DNA fragments. This is also true for Chd1 (*McKnight et al., 2011*; *Stockdale et al., 2006*). Repositioning of nucleosomes with this directionality conflicts with the directionality of translocation inferred from docking the tracking strand of NS3 into Chd1 (*Farnung et al., 2017*) (*Figure 6*). Tracking along this strand with 3'−5' directionality would instead be anticipated to draw DNA into the nucleosome from the side of the nucleosome that has no linker DNA.

Inferring the mechanism of Chd1 from NS3 is complicated by the fact these enzymes are not so closely related. Conserved motifs are difficult to align based on sequence alone. In addition some aspects of nucleic acid binding by both Snf2 and Chd1 profoundly differ from NS3. Notably, motifs II and III within lobe one contact the opposite, 3'−5' stand, which is not present in NS3. In addition, Snf2-related chromatin remodelling enzymes contain features that extend the nucleic acid binding cleft between the two ATPase lobes and make contacts with both strands (*Figure 6*). As a result of the extensive contacts with both strands, it is possible that the assignment of guide and tracking strands within remodelling ATPases is not absolute as tracking may be coupled to both strands. Consistent with this, experiments that have probed the action of remodelling enzymes using short gaps in either strand of nucleosomal DNA have found them to be sensitive to lesions in either strand (*Saha et al., 2005*; *Zofall et al., 2006*).

The introduction of gaps in nucleosomal DNA has also been used to infer the directionality with which ATPases's move along DNA. Introduction of gaps distal to the SHL two location closest to the entry linker DNA has been observed to impede the action of Snf2, Iswi and Chd1 enzymes (*McKnight et al., 2011*; *Saha et al., 2005*; *Zofall et al., 2006*). This has been used as evidence that the enzyme translocation that drives repositioning initiates from the SHL two located distal to the entry DNA. As a consequence it has been proposed that Chd1 bound in the cross gyres conformation, that we and others observe, represents an inactive state in which the interaction of the DNABD with (exit) linker DNA is inhibitory (*Nodelman et al., 2017*).

Confounding this, the Chd1-nucleosome structure bears the hallmarks of an active DNA translocase. Density for ADP.BeF is observed in the nucleotide binding pocket, and the ATPase domains are repositioned to a closed conformation with conserved residues positioned for catalysis (*Figure 2—figure supplement 3*). The closure of the ATPase domains in the Chd1-nucleosome complex is connected to repositioning of the chromodomains, which in turn levers the DNABD position and allows DNA to gain access to the cleft between the ATPase domains.

Further study will be required to reconcile these observations. We nonetheless speculate that the initial stages of remodelling by Chd1 are most effective on nucleosomes lacking exit DNA. Consistent with this the initial rate of Chd1 ATPase activity has been observed to be greater on nucleosomes lacking exit linker DNA (*Nodelman et al., 2017*). In cells, lack of linker could arise in close packed nucleosomes and represent a suitable substrate for a nucleosome spacing enzyme. The rapid action of Chd1 on close packed chromatin would generate new exit linker DNA which would then be available to be bound by the DNABD in the conformation observed by in the EM structures. In this case, Chd1 bound in the conformation shown in *Figure 1*, could be active but represent a state in which the DNABD is bound to exit rather than entry DNA. This reconciles many of the biochemical observations, and is compatible with the directionality inferred from the interaction of DNA with the ATPase domains. However, it requires that the Chd1 ATPase domains exhibit a preference for initial binding to nucleosomes on the side that initially lacks linker DNA. The basis for this is not clear, but in the case of ISWI enzymes the initial and subsequent translocation events have been observed to differ (*Deindl et al., 2013*), raising the possibility that they could be regulated independently. Following initial repositioning, the engagement of the DNABD with the new exit linker may provide an opportunity for large changes in enzyme conformation, perhaps related to those observed for SNF2H (*Leonard and Narlikar, 2015*) and enabling individual Chd1 molecules to switch between different sides of the nucleosome enabling bidirectional repositioning (*Qiu et al., 2017*).

## Interplay between Chd1 and histone modifications

Both budding yeast Chd1 and human Chd2 are found to be enriched within coding regions (*de Dieuleveult et al., 2016*; *Gkikopoulos et al., 2011*; *Lee et al., 2017*). Histone H3 K36me3 is a hallmark of coding region nucleosomes, so we prepared nucleosomes alkylated to mimic trimethylation at this position. Alkylation modestly stimulates Chd1 activity (*Figure 1—figure supplement 1*), raising the possibility that this modification is recognised by the enzyme, possibly via the chromodomains. However, we observe electron density for the histone H3 tail to residue 26, indicating that H3K36 does not stably interact with the chromodomains or any other component of Chd1 in the structure reported here. Furthermore, for this interaction to occur, either the chromodomains would need to be repositioned, or the structure of the N-terminus of H3 reconfigured for example by unfolding of the alpha–N helix (*Elsässer et al., 2012*; *Liu et al., 2012*).

The improved density for the H3 tail on the unwrapped side of the nucleosome is most likely to result from the interaction of the basic N-terminal region of the H3 tail, which is not resolved, with DNA. It is notable that in the fully wrapped state the H3 tail would need to follow a very different path in order to interact with DNA. Consistent with this the trajectory of the H3 tail on the unwrapped side of the nucleosome is different to that observed in structures of intact nucleosomes (*Wilson et al., 2016*). This raises the possibility that changes to DNA wrapping could affect the way in which histone tail epitopes are displayed. In principle, such effects could be positive or negative. For example the tudor domain of PHF1 preferentially interacts with trimethylated H3K36 on partially unwrapped nucleosomes (*Gibson et al., 2017*). The interaction of the PHD domains of Chd4 with DNA is also inhibited by nucleosomal DNA (*Gatchalian et al., 2017*). As a result if Chd4 generates unwrapped structures similar to those observed with Chd1 the interaction of these domains would be enhanced. The reconfiguration of the H3 tail by Chd1 has the potential to affect the interaction of histone reader, writer and eraser enzymes with the tail and as a result the distribution of these modifications in chromatin. Such effects have been observed, as Chd1 contributes to the establishment of boundaries between H3K4me3 and H3K36me3 at most transcribed genes (*Lee et al., 2017*).

H2BK120ub is also enriched in coding region chromatin, and directly stimulates Chd1 activity (*Levendosky et al., 2016*) (*Figure 8—figure supplement 2*) (*Sundaramoorthy et al., 2017*). It has also been observed that H2BK120ub negatively affects the activity of some ISWI containing enzymes (*Fierz et al., 2011*). As organisation of coding region nucleosomes involves these and other enzymes

(*Krietenstein et al., 2016*; *Ocampo et al., 2016*; *Parnell et al., 2015*), H2BK120Ub has the potential to regulate interplay between different enzymes.

Ubiquitin on the unwrapped side of the nucleosome is repositioned such that it interacts directly with DNA. As in the case of the H3 tail, the repositioning of the ubiquitin resulting from Chd1-directed DNA unwrapping could potentially affect interactions with the factors involved in the placing, removal or recognition of H2BK120ub. The most striking functional evidence for this interplay is that H2BK120ub is greatly reduced in Chd1 mutants (*Lee et al., 2012*). One possible explanation for this effect is that Chd1 sequesters ubiquitin in a conformation less accessible for removal. Consistent with this, the position of ubiquitin on the unwrapped side of Chd1 bound nucleosomes is incompatible with interaction with the SAGA DUB module (*Morgan et al., 2016*). Interestingly, the paradigm for trans regulation between histone modifications stems from the interplay between H2BK120ub and H3 K4 methylation (*Sun and Allis, 2002*), both of which are influenced by Chd1 binding. While Chd1 is not required for H3K4me3 (*Lee et al., 2012*), it does influence the distribution of this histone modification (*Lee et al., 2017*).

H2BK120ub has previously been observed to directly affect chromosome structure at the level of chromatin fibre formation (*Debelouchina et al., 2017*; *Fierz et al., 2011*). Our observations show a new role for H2BK120ub at the level of nucleosomal DNA wrapping. The specific relocation of ubiquitin on the unravelled side of the nucleosome, the local distortion of H2B at the site of attachment and the presence of lysine and arginine residues at the site of interaction with DNA all indicate this is a favourable interaction that stabilises DNA in the unwrapped state. The outer turns of nucleosomal DNA rapidly associate and dissociate on millisecond time scales, with occupancy of the unwrapped state estimated at 10% (*Li et al., 2005*). The ubiquitin interaction we have observed would be anticipated to stabilise the transiently unwrapped state increasing its abundance. It is however, unlikely that the unwrapped state predominates in the absence of Chd1 or other factors that promote unwrapping as the structure of isolated ubiquitinylated nucleosomes is unchanged (*Machida et al., 2016*). Nonetheless, increased occupancy of the transiently unwrapped state would be anticipated to facilitate access to nucleosomal DNA. Chromatin folding to form higher order structures is likely to be favoured by fully wrapped nucleosomes, and so an increase in the proportion of unwrapped nucleosomes could potentially contribute to the effects of H2BK120ub on chromatin fibre formation (*Fierz et al., 2011*). Many other processes involving chromatin dynamics are linked to H2BK120ub including transcription (*Bonnet et al., 2014*), DNA repair (*Moyal et al., 2011*; *Nakamura et al., 2011*) and DNA replication (*Lin et al., 2014*). A more stable unwrapped state could also provide an explanation for the association of factors that lack recognised ubiquitin interaction domains, with ubiquitinylated chromatin (*Shema-Yaacoby et al., 2013*). Interestingly, H2BK120ub associating proteins include human Chd1, SWI/SNF complex, pol II and the elongation factors NELF and DISF (*Shema-Yaacoby et al., 2013*).

The change in the position of ubiquitin also has the potential to indirectly affect the way in which other factors interact with ubiquitinylated nucleosomes. On the wrapped side of the nucleosome, ubiquitin is positioned such that it occludes access to the acidic patch formed by the cleft between histones H2A and H2B. This provides a surface via which many proteins including LANA peptides (*Barbera et al., 2006*), RCC1 (*Makde et al., 2010*), Sir3 (*Armache et al., 2011*), PRC1 (*McGinty et al., 2014*) and the SAGA DUB module (*Morgan et al., 2016*) interact with nucleosomes. The repositioning of ubiquitin away from the acidic patch on the unwrapped side of the nucleosome improves access to the acidic patch. In this way, H2BK120ub may provide a means of regulating access to the acidic patch that is sensitive to changes in nucleosome structure.

Although the repositioning of the H3 tail and ubiquitin were observed on Chd1-bound nucleosomes, the potential for reconfiguration of histone epitopes may be more general. All processes that generate local DNA unwrapping would be anticipated to result in similar repositioning of histone tail epitopes. In particular, where combinations of modifications are recognised bivalently, the spatial alignment of epitopes will be important for recognition by coupled reader domains. This potentially provides a means of tuning signalling via histone modifications to regions of transient histone dynamics.

## Materials and methods

### Cloning, protein expression and purification

ScChd1 C-terminal and N-terminal truncations were made from the full length clone described in Ryan et al, using an inverse PCR strategy (*Ryan et al., 2011*). A similar approach was used to generate a chd1 lobe2 Δ632–646 deletion. Site-directed mutagenesis was used to introduce cysteine residues at strategic locations on ScChd1 1-1305ΔC using standard cloning procedure. All proteins were expressed in Rosetta2 (DE3) pLysS *Escherichia coli* cells at 20° C in Auto-induction media, and the purification of the protein was carried out typically as described in Ryan et al. After the purification of the protein the GST tag was cleaved with precision protease and the tag cleaved proteins were subjected to size exclusion chromatography using Superdex S200 10/300 GL columns (GE Healthcare). Expression and purification of Xenopus laevis histones were carried out as described previously (*Luger et al., 1999*).

### Installation of Methyl-lysine analogues in H3 K36

Alkylation of cysteine-mutant histones to generate histones modified with methyl-lysine analogues was performed as in (*Simon et al., 2007*). Approximately 10 mg of lyophilised cysteine mutant histone was resuspended in 800 µL (me3) or 900 uL (me0) degassed alkylation buffer (1M HEPES, 10 mM D,L-methionine, 4M Guanidine HCl, pH7.8). Histones were reduced with fresh 30 mM DTT for 30 min at room temperature.

For trimethyl-lysine analogues, the reduced histone was added to approximately 125 mg of (2-Bromoethyl) trimethylammonium bromide (Sigma 117196–25G) in 200 µL of DMF and incubated in the dark at $50^0$C for 3 hr. An additional 10 µL of DTT was added, and the reaction was allowed to proceed overnight at room temperature.

For generation of the unmethylated lysine analogue, 75 µL of 1M 2-Bromoethylamine hydrobromide (Fluka 06670–100G) was added to the reduced histone and was incubated at room temperature in the dark for 3 hr. An additional 10 µL of DTT was added for 30 min prior to the addition of an extra 75 µL of alkylating agent, and the reaction was allowed to proceed overnight at room temperature in the dark.

The reaction was terminated with the addition of 50 µL 2-mercaptoethanol for 30 min and the alkylated histone was desalted either by dialysis into water with 2 mM 2-mercaptoethanol or on a PD-10 desalting column (GE 52130800). The shift in molecular weight associated was confirmed via MALDI-TOF mass spectrometry.

### In vitro ubiquitination

Recombinant expression of xH2B K120C and His-TEV-Ubiquitin G76C mutant proteins was induced with IPTG for 4 hr in Rosetta 2 DE3 pLysS cells grown at $37^0$C. Inclusion body purification followed by cation exchange chromatography was performed to isolate the histone protein. Ubiquitin was purified using HisPur cobalt resin with 150 mM sodium chloride/20 mM Tris pH8 buffer and eluted with 350 mM imidazole. Histones and ubiquitin were desalted by dialysis into water with 2 mM 2-mercaptoethanol and lyophilised for storage.

Proteins were re-suspended in 50 mM ammonium bicarbonate pH eight and treated with 2 mM TCEP for 1 hr. Ellman's reagent was used to ascertain the concentration of free sulfhydryls, and xH2b and ubiquitin were combined at equimolar ratios, as defined by the Ellman's assay, and diluted with 50 mM ammonium bicarbonate to 200–250 uM each protein. The proteins were crosslinked at room temperature with four hourly additions of 1/3 molar ratio of 1,3 dichloroacetone (freshly prepared in DMF). An equal volume of denaturing buffer (7M Guanidine HCl, 350 mM sodium chloride, 25 mM Tris pH7.5) was added to the reactions, which were purified using HisPur cobalt resin, pre-equilibrated in denaturing buffer. The His-TEV-Ub-xH2B crosslinked product was eluted with 350 mM imidazole and dialysed into SAUDE200 buffer (7M Urea, 20 mM sodium acetate, 200 mM sodium chloride, 1 mM EDTA, 5 mM 2-mercaptoethanol) overnight. The ubiquitinated histone was further purified over a cation exchange column, as before, and fractions were dialysed into water with 2 mM 2-mercaptoethanol and lyophilised for storage.

## Preparation of recombinant nucleosomes

Xenopus H2B-K120 ubiquitinylated histones were refolded in equimolar ratios with H2A and similarly H3 K36 methyl analogue histones were refolded in equimolar ratios with histone H4 to obtain dimers and tetramers as described previously for wild type histones Dyer et al., and purified on a size exclusion chromatography using S200 gel filtation column. The peak fractions were analysed with SDS-PAGE gel and pooled. 601 DNA fragments of respective lengths for recombinant nucleosome assembly were generated by PCR method as described previously (*Sundaramoorthy et al., 2017*). Nucleosomes were generated by salt dialysis as described previously by combining H2A/H2B-K120 ubiquitin dimer, H3K36 methyl lysine analogue tetramer (2:1 ratio) with DNA containing PCR-amplified Widom 601 DNA sequence.

## Nucleosome repositioning assay

Nucleosomes were reconstituted on Cy3 (me0) and Cy5 (me3) labelled DNA, based on the 601 sequence, with a 47 bp extension. Repositioning by Chd1 was performed in 40 mM Tris pH7.4, 50 mM KCl, 3 mM MgCl2, 1 mM ATP, 100 nM each nucleosome, and 10 nM Chd1; 10 µL was removed at each time point (T = 0, 4, 8, 16, 32, and 64 min), placed on ice, and stopped with the addition of 100 ng/µL competitor DNA, 200 mM NaCl, and 1.6% sucrose. Repositioned nucleosomes were run on 6% PAGE/0.2X TBE gels in recirculating 0.2X TBE buffer for 3–4 hr at 300V. The percent of repositioned nucleosomes was analysed using Aida image analysis software. Data were fit to a hyperbola in Sigma Plot, to determine the initial rate of repositioning.

## Nucleosome binding

*Xenopus laevis* nucleosomes (20 nM), reconstituted on Cy3 labelled 0W11 DNA, were bound to titrations of Chd1 enzymes (concentration specified in figure legend) in 50 mM Tris pH 7.5, 50 mM sodium chloride, and 3 mM magnesium chloride supplemented with 100 µg/mL BSA. Unbound and bound nucleosomes were separated on a pre-run 6% polyacrylamide gel (49:1 acrylamide: bis-acrylamide) in 0.5X TBE buffer for 1 hr at 150V. The gel shift was scanned on Fujifilm FLA-5100 imaging system at 532 nm.

## Spin labelling of ScChd1, PELDOR measurements and modelling

MTSL was conjugated to introduced cysteines immediately following size exclusion purification as described in *Hammond et al. (2016)*. Excess unreacted labels were removed from the sample by dialysis. PELDOR experiments were conducted at Q-band (34 GHz) operating on a Bruker ELEXSYS E580 spectrometer with a probe head supporting a cylindrical resonator ER 5106QT-2w and a Bruker 400 U second microwave source unit as described previously (*Hammond et al., 2016*). All measurements reported here were made at 50K. Data analysis was carried out using the DeerAnalysis 2013 package (*Jeschke and Polyhach, 2007*). The dipolar coupling evolution data were first corrected to remove background decay. Tikhonov regularisation was then used to determine distance distributions from each dataset.

To model the distance distribution for the open conformation of Chd1 helicase lobes crystal structure of chromo helicase (PDB Code: 3MWY) (*Hauk et al., 2010*) was used. For the closed conformation refined cryoEM structure of Chd1 bound to nucleosome in the presence of ADP.BeF$_x$ described in this study was used as a model. For each structure, R1 spin labels were added and the distribution simulated for each position using MTSSL wizard in Pymol. Also the average distance from the distribution from a pair of spin labels were calculated using MTSSL wizard in Pymol.

## Sample preparation, Cryo Electron Microscopy data collection and analysis

The appropriate ratio of ScChd1(1-1305Δ57–88) to nucleosome for 1:1 and the 2:1 complex formation in the presence of 5-fold molar excess of ADP-BeF$_x$ was determined by titration and native PAGE analysis. The formed complex was then purified by size exclusion gel filtration using a PC 3.2/30 superdex 200 analytical column in 20 mM Tris, 50 mM NaCl and 250 µM ADP.BeF$_x$.(formed by mixing 1:1:3 molar ratio of ADP:BeCl$_2$:NaF). In a typical run 50uLs of 20 µM of sample was injected using Dionex autoloader. 50uLs fractions were collected and analysed in 6% Native PAGE gel and appropriate fractions containing ScChd1-nucleosome complexes were pooled together. A 4 µl drop

of sample was then applied to C-flat Holey carbon foil (400 mesh R1.2/1.3 µM) pre-cleaned with glow discharge (Quorum technologies). After 15 s incubation, grids were double side blotted for 4 s in a FEI cryo-plunger (FEI Mark III) at 90% humidity and plunge frozen into −172°C liquefied ethane. Standard vitrobot filter paper Ø 55/20 mm, Grade 595 was used for blotting.

The prepared grids are initially checked for its ice quality and the particle distribution using a JEOL 2010 microscope with side-entry cryo-holder operated at 200 keV and equipped with a gatan 4k × 4 k CCD camera. Cryo-grids were then stored in liquid nitrogen and dry-shipped to respective centre for data collection. For the 1:1 complex the data were acquired on a FEI Titan Krios transmission electron microscope (TEM) operated at 300 keV, equipped with a K2 summit direct detector (Gatan). The 2:1 complex data were collected with FEI Titan Krios microscope equipped with Falcon three detector. Automated data acquisition was carried out using FEI EPU software at a nominal magnification of 105,000 × for the 1:1 complex and 59,000X for the 2:1 complex. For both the datasets, the data were collected as a movie with 32 frames per movie for the 1:1 complex and 40 frames per movie for the 2:1. Details of the data collection and processing parameters for both the datasets are included in *Table 1*.

The movie frames were subjected to frame wise motion correction using MotionCor2 (*Zheng et al., 2017*). CTF correction was then performed on the motion corrected summed image using GCTF (*Zhang, 2016*). Subsequent image processing was performed with RELION 2.0.4 (*Scheres, 2012*). About 5000 particles from 50 micrographs were first handpicked in RELION, extracted and 2D classes were generated. These 2D classes were then used as a reference in RELION auto pick routine and particles were picked from respective number of micrographs from each dataset. The auto picked particles were subsequently extracted and sorted. An iterative round of two-dimensional classification was performed to discard poorly averaging particles, contamination and exploded particles. On the resultant cleaned up particle stack a hierarchical three-dimensional classification and refinement was performed as described in the results. A low pass filtered low resolution chd1 engaged nucleosome structure was used as an initial model in the 3D classification. At the three-dimensional refinement stages a soft mask that encompasses the entire Chd1-nucleosome complex was applied. Post processing of refined models was performed with automatic B factor determination in RELION. Resolution for the refined reconstruction are determined at 0.143 FSC cut off as 4.5 Å for the 1:1 complex and 10 Å for the 2:1 Complex. Local resolution estimates were determined for the 1:1 complex using Resmap-1.4.

**Table 1.** Data collection and processing summary

| Data collection and processing | UbNucleosome-Chd1 complex | Nucleosome-2Chd1 complex |
|---|---|---|
| Microscope | Titan Krios | Titan Krios |
| Detector | Gatan K2 Summit | Falcon 3 |
| Mode | Electron Counting | Integrating |
| Magnification | 105000X | 59000X |
| Voltage (kV) | 300 | 300 |
| Number of frames | 32 | 40 |
| Electron exposure (e⁻/Å²) | 40 | 62.5 |
| Exposure time (Sec) | 10 | 1.5 |
| Dose per fraction | 1.25 | 1.56 |
| Pixel Size | 1.4 | 1.42 |
| Defocus Range | 1.5–3.5 | 1.7–3.5 |
| Symmetry Imposed | C1 | C1 |
| Initial Particle images (no.) | 1050000 | 275504 |
| Final Particle images (no.) | 137,000 | 24846 |
| Final Map resolution (Å) | 4.5 | 10 |
| Map Sharpening B factor (Å²) | −200 | −800 |

DOI: https://doi.org/10.7554/eLife.35720.034

## Model building

For model building X.laevis nucleosome with Widom 601 sequence (PDB 3LZ0), the S.cerevisiae Chd1 DNA-binding domain (PDB 3TED) and the ATPase core with tandem chromo domain (3MWY) were used. The domains were individually placed into the electron density using UCSF chimera and fitted as a rigid body. The path of the unwrapped DNA, H4 tail region and H3 tail region were manually built in Coot. Protein back bone restraints and DNA base pair, and parallel pair restraints were generated using ProSMART and LibG modules (*Nicholls et al., 2012*). The generated restraints were then used as constraint in jelly body refinement with CCPEM REFMAC program. ADP·BeF$_3$ was built by superpositioning ATP-gamma-S from the inactive Chd1 structure (PDB code 3MWY) onto our model, inspected in COOT and replacing the ATP analogue with ADP·BeF$_x$. For the 2:1 complex the structural model of 2chd1 bound to one nucleosome with 14 bp symmetrical linker on both side was generated using one chd1 bound to 0W14 nucleosome presented in this study and the one chd1 bound to 63W0 nucleosome (PDB-ID 5O9G). The model was then rigid body docked into the reconstructed 2chd1:1nucleosome volume in chimera. The quality of the fit is assessed by visual examination and the correlation coefficient between the model and the map. Sequence alignments were generated using JALVIEW (*Waterhouse et al., 2009*). The EM volumes and the fitted model can be accessed from the EMDB database with EMDB accession number EMD-4318 and PDB code 6FTX for the 1:1 complex and EMD-4336 and EMD-4336 for the 2:1 complex.

## Acknowledgements

We thank Daniel Claire electron Bio-Imaging Centre (eBIC), Diamond light source Ltd, UK for data collection with respect to the 4.5 Å structure. We thank Rebecca Thompson and Neil Ranson for assistance with collection of 2Chd1 bound to one nucleosome structure at the Astbury Biostructre Laboratory, University of Leeds. The ubiquitin expression plasmid was kindly provided by Ron Hay, University of Dundee. This work was funded by Wellcome Senior Fellowship 095062, Wellcome Trust grants 094090, 099149 and 097945. ALH was funded by an EMBO long term fellowship ALTF 380–2015 co-funded by the European Commission (LTFCOFUND2013, GA-2013–609409). We thank Dale Wigley and Chris Aylett for useful discussions.

## Additional information

### Funding

| Funder | Grant reference number | Author |
|---|---|---|
| Wellcome | 095062 | Ramasubramanian Sundaramoorthy Amanda L Hughes Hassane El-Mkami David G Norman Tom Owen-Hughes |
| European Molecular Biology Organization | ALTF 380-2015 | Amanda L Hughes |
| Wellcome | 094090 | Ramasubramanian Sundaramoorthy Amanda L Hughes Hassane El-Mkami David G Norman Tom Owen-Hughes |
| Wellcome | 099149 | Ramasubramanian Sundaramoorthy Amanda L Hughes Hassane El-Mkami David G Norman Tom Owen-Hughes |

| Wellcome | 097945 | Ramasubramanian Sundaramoorthy<br>Amanda L Hughes<br>Hassane El-Mkami<br>David G Norman<br>Tom Owen-Hughes |
|---|---|---|

The funders had no role in study design, data collection and interpretation, or the decision to submit the work for publication.

## Author contributions

Ramasubramanian Sundaramoorthy, Conceptualization, Data curation, Formal analysis, Investigation, Methodology, Writing—original draft, Project administration, Writing—review and editing; Amanda L Hughes, Conceptualization, Investigation, Methodology, Writing—original draft, Project administration, Writing—review and editing; Hassane El-Mkami, Investigation, Methodology, Writing—original draft, Writing—review and editing; David G Norman, Conceptualization, Formal analysis, Funding acquisition, Writing—original draft, Writing—review and editing; Helder Ferreira, Conceptualization, Formal analysis, Investigation, Methodology, Writing—review and editing; Tom Owen-Hughes, Conceptualization, Supervision, Funding acquisition, Investigation, Writing—original draft, Project administration, Writing—review and editing

## Author ORCIDs

Ramasubramanian Sundaramoorthy https://orcid.org/0000-0003-4895-0980
Amanda L Hughes http://orcid.org/0000-0002-1976-6634
David G Norman http://orcid.org/0000-0002-7658-7720
Tom Owen-Hughes http://orcid.org/0000-0002-0618-8185

## Decision letter and Author response

Decision letter https://doi.org/10.7554/eLife.35720.046
Author response https://doi.org/10.7554/eLife.35720.047

# Additional files

## Supplementary files

• Supplementary file 1. Primers used in generating the deletion Δ632–646 on Chd1. DNA sequence of Chd1 Δ632–646. Protein sequence of Chd1 Δ632–646. Primers used to generate H3D1-25. Protein sequence of H3D1-25. DNA sequence of H3D1-25. Protein sequence of H3D1-37. DNA sequence of H3D1-37.
DOI: https://doi.org/10.7554/eLife.35720.035

• Transparent reporting form
DOI: https://doi.org/10.7554/eLife.35720.036

## Data availability

The EM structures generated have been deposited in EMDB and PDB under the accession codes EMD-4318, PDB 6FTX, EMD-4336 and PDB 6G0L.

The following datasets were generated:

| Author(s) | Year | Dataset title | Dataset URL | Database, license, and accessibility information |
|---|---|---|---|---|
| Owen-Hughes S | 2018 | Structure of the chromatin remodelling enzyme Chd1 bound to a ubiquitinylated nucleosome | http://www.ebi.ac.uk/pdbe/entry/emdb/EMD-4318 | Publicly available at the Electron Microscopy Data Bank (accession no. EMD-4318) |
| Owen-Hughes S | 2018 | Structure of the chromatin remodelling enzyme Chd1 bound to a ubiquitinylated nucleosome | http://www.rcsb.org/structure/6FTX | Publicly available at the RCSB Protein Data Bank (accession no. 6FTX) |

| Owen-Hughes S | 2018 | Structure of two molecules of the chromatin remodelling enzyme Chd1 bound to a nucleosome | http://www.ebi.ac.uk/pdbe/entry/emdb/EMD-4336 | Publicly available at the Electron Microscopy Data Bank (accession no. EMD-4336) |
|---|---|---|---|---|
| Owen-Hughes S | 2018 | Structure of two molecules of the chromatin remodelling enzyme Chd1 bound to a nucleosome | http://www.rcsb.org/structure/6G0L | Publicly available at the RCSB Protein Data Bank (accession no. 6G0L) |

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
