## [Decision Letter]

Thank you for submitting your article "Structure of the chromatin remodelling enzyme Chd1 bound to a ubiquitinylated nucleosome" for consideration by *eLife*. Your article has been reviewed by three peer reviewers, including Geeta J Narlikar as the Reviewing Editor and Reviewer #1, and the evaluation has been overseen by John Kuriyan as the Senior Editor. The following individual involved in review of your submission has agreed to reveal their identity: Song Tan (Reviewer #3).

The reviewers have discussed the reviews with one another and the Reviewing Editor has drafted this decision to help you prepare a revised submission.

Summary:

The Chd1 chromatin remodeler is important for nucleosome organization at promoters and coding regions in vivo and its presence is positively correlated with the coding region histone marks, H2BK120-Ubiquitylation and H3K36-methylation. In vitro Chd1 has been shown to have nucleosome assembly and nucleosome sliding activities. The link between these in vitro and in vivo activities is an area of active study and it is not known which of the in vivo effects of Cdh1 are direct vs. indirect. Sundaramoorthy et al. present a cryo-EM structure of Chd1 bound to a nucleosome with ubiquitylation on H2BK120 and tri-methylation mark on H3K36 in the presence of ADP-BeF_x_, at 4.5 Å resolution. In addition the authors obtain lower resolution reconstructions of doubly bound Chd1 molecules on a mono-nucleosomes.

The singly bound structure largely resembles the published Farnung et al. structure obtained with an unmodified mono-nucleosome. Despite the overall similarities the reviewers felt the Sundaramoorthy et al. study provided new insights that would be of much interest to researchers in the field of chromatin regulation. Firstly, the results suggest an interesting new mechanism for how ubiquitylation of histone H2B Lys120 could affect chromatin biology without requiring a specific reader protein for this chromatin modification. The structure is consistent with a model in which Chd1 action promotes H2B-ubiquitylation by protecting it from de-ubiquitylation and promotes H3 modifications by relieving inhibitory tail-DNA contacts. Secondly, the manuscript provides substantially more information and richer analysis than the Farnung et al. Nature Letter. A particularly important example is the discussion of the directionality of Chd1-mediated nucleosome repositioning. Both structures appear to be inconsistent with the observed direction of translocation, but as Sundaramoorthy et al. discuss, this analysis is based on comparisons with the NS3 single-stranded DNA translocase and it is not clear that the NS3 translocase is a good model for double-stranded DNA translocases like Chd1. Thirdly, unlike the published work this structure is obtained with using only Chd1 and nucleosomes.

While overall the reviewers feel this work could be suitable for *eLife*, there are some major concerns that the reviewers would like to see addressed before acceptance as outlined below.

Essential revisions:

1) For Figure 1, there are some concerns about whether the resolution of the 3D reconstruction is overestimated. Normally, in maps with resolution higher than 5 Å, bulky side chains become visible and β strands become separated, but such features cannot be recognized from the reconstruction, even in Figure 1—figure supplement 5. The local resolution estimation for the core histones is also shown to be ~4 Å resolution in Figure 1—figure supplement 4, however it appears lower than 4 Å. One way the authors can address this concern idea by using the high-resolution noise substitution method (Chen et al., Ultramicroscopy, 2013) in RELION. The authors should compare among the FSC curves of the masked FSC, the unmasked FSC, the phase-randomized FSC, and the masking-effect-corrected FSC.

2) The authors need to show multiple views of the final reconstructions, because Figure 1—figure supplement 4B is the only view from the top. It is easier to understand the 3D structure, if multiple views are presented. In particular, locations of the ubiquitin at the unwrapped and wrapped sides of the nucleosome are important points in this manuscript, both of which are missing in this figure.

3) Regarding local resolution estimation in Figure 1—figure supplement 4B, the authors mentioned that "The resolution varies within the map, with resolution close to 4 Å in the region occupied by the nucleosome and ATPase lobes and lower resolution in the vicinity of the DNABD and ubiquitin peptides (Figure 1—figure supplement 4B)." However, I cannot recognize the ubiquitin peptides in the map with local resolution. The authors need to change the contour level of the map to allow visualization of the ubiquitin peptides on the local resolution estimation map.

4) The FSC curve of the 2:1 Chd1-nucleosome complex (Figure 2—figure supplement 1E, no indication for X-Y axis) does not appear healthy enough to determine the resolution of the 3D reconstruction. According to the book chapter written by Pawel Penczek (Resolution Measures in Molecular Electron Microscopy, Methods in Enzymology, p73-p100, 2010), "Artifactual "rectangular" FSC: remains one at low frequencies, followed by a sharp drop, in high frequencies oscillates around zero. Typically, it is caused by a combination of alignment of noise and a sharp filtration during the alignment procedure. The FSC never drops to zero in the entire frequency range. Normally, this means that the noise component in the data was aligned, the results are artifactual and the resolution is undetermined." To avoid the artifactual reconstruction from noise, as performed by Mao et al. (PNAS, 2013; also see Henderson, PNAS, 2013), for example, the authors need to reprocess the data to obtain a reliable FSC for the 3D reconstruction.

5) There were a few concerns about the reasoning used by the authors to suggest that singly bound structure is the active conformation.

i) The authors do not explain how the published DNA gaps data can be reconciled with the cross-gyres conformation being the active conformation as opposed to the same-gyres conformation.

ii) Previous work has shown that Chd1 can move nucleosomes by 23-29 base-pairs without a stall, which is more than the 7 bp that is contacted by the DBD. The authors use this data as evidence against the cross-gyres conformation. This reasoning is not clear because the ISWI ATPases (Deindl et al., 2013) have step sizes as assessed by smFRET that are not correlated with the amount of DNA that is contacted by their DBDs. Further depending on the processivity of the enzyme the amount of DNA moved before a stall could be several multiples of 7 bp.

iii) In the Discussion the authors say: "This is anticipated if the DNA binding domain acts to generate an active conformation, but not if sensing of exit linker DNA is repressive." However, the ATPase data from Nodelman et al. (2017) shows a repressive effect of exit linker DNA. This should be discussed.

iv) In the Discussion the authors say: "Thirdly, the activity of chimeric Chd1 proteins in which DNA binding is provided via a heterologous domain is greatest when the cognate binding site is placed in the entry linker mimicking the arrangement observed in the Chd1-nuceosome structure". While this data is consistent with the cross-gyres conformation being the active conformation, the data is also consistent with the same-gyres conformation being active if what is being trapped by EM is the inactive state. In other words this prior data does not rule out the same-gyres conformation being the active conformation.

6) A few straightforward biochemical tests are required to test the mechanistic significance of the structural data. The authors can investigate the functional effects on nucleosome sliding of mutating the Chd1 lobe II residues that make contacts with the H3 alpha 1 helix and the 25 N-terminal residues that are hypothesized to contact the unpeeled DNA.

---

## [Author Response]

Essential revisions:1) For Figure 1, there are some concerns about whether the resolution of the 3D reconstruction is overestimated. Normally, in maps with resolution higher than 5 Å, bulky side chains become visible and β strands become separated, but such features cannot be recognized from the reconstruction, even in Figure 1—figure supplement 5. The local resolution estimation for the core histones is also shown to be ~4 Å resolution in Figure 1—figure supplement 4, however it appears lower than 4 Å. One way the authors can address this concern idea by using the high-resolution noise substitution method (Chen et al., Ultramicroscopy, 2013) in RELION. The authors should compare among the FSC curves of the masked FSC, the unmasked FSC, the phase-randomized FSC, and the masking-effect-corrected FSC.

FSC curves generated using the high resolution noise substitution method confirm our original assessment of overall average resolution as 4.5 Å (new Figure 1—figure supplement 4A). There is a masking effect and the unmasked FSC is lower at 6.5 Å (new Figure 1—figure supplement 4A). The phase randomised FSC indicates there is no overfitting. We removed the text “close to 4Å” to avoid the potential for over interpreting areas of higher local resolution. The colour scheme in the original Figure 1—figure supplement 4 did not clearly distinguish resolution in the range 4-5 Å, the revised version includes an improved colour map of the unfiltered envelope with and additional view. We have included an example illustrating that some bulky side chains fit the unsmoothed density map (new Figure 1—figure supplement 5). This is comparable to the fit observed in the 4.5 Å structure of 53BP1 bound to a nucleosome (PDB:5KGF) and not as good as that observed in the 3.9 Å structure (EMDB 8140).

2) The authors need to show multiple views of the final reconstructions, because Figure 1—figure supplement 4B is the only view from the top. It is easier to understand the 3D structure, if multiple views are presented. In particular, locations of the ubiquitin at the unwrapped and wrapped sides of the nucleosome are important points in this manuscript, both of which are missing in this figure.

Additional views are now presented in the new Figure 1—figure supplement 4B.

3) Regarding local resolution estimation in Figure 1—figure supplement 4B, the authors mentioned that "The resolution varies within the map, with resolution close to 4 Å in the region occupied by the nucleosome and ATPase lobes and lower resolution in the vicinity of the DNABD and ubiquitin peptides (Figure 1—figure supplement 4B)." However, I cannot recognize the ubiquitin peptides in the map with local resolution. The authors need to change the contour level of the map to allow visualization of the ubiquitin peptides on the local resolution estimation map.

Density for the ubiquitin peptides is visible in the revised figure.

4) The FSC curve of the 2:1 Chd1-nucleosome complex (Figure 2—figure supplement 1E, no indication for X-Y axis) does not appear healthy enough to determine the resolution of the 3D reconstruction. According to the book chapter written by Pawel Penczek (Resolution Measures in Molecular Electron Microscopy, Methods in Enzymology, p73-p100, 2010), "Artifactual "rectangular" FSC: remains one at low frequencies, followed by a sharp drop, in high frequencies oscillates around zero. Typically, it is caused by a combination of alignment of noise and a sharp filtration during the alignment procedure. The FSC never drops to zero in the entire frequency range. Normally, this means that the noise component in the data was aligned, the results are artifactual and the resolution is undetermined." To avoid the artifactual reconstruction from noise, as performed by Mao et al. (PNAS, 2013; also see Henderson, PNAS, 2013), for example, the authors need to reprocess the data to obtain a reliable FSC for the 3D reconstruction.

We appreciate the concerns raised. The data has been reprocessed as suggested. The revised FSC curves are much healthier allowing resolution to be estimated at 11 Å.

5) There were a few concerns about the reasoning used by the authors to suggest that singly bound structure is the active conformation.

The structures of Chd1 bound nucleosomes present a conundrum as the directionality inferred from the structures is opposite to what is anticipated based on the known biochemical properties of Chd1. The comments below have been thought provoking and we have revised and shortened this part of the discussion substantially.

i) The authors do not explain how the published DNA gaps data can be reconciled with the cross-gyres conformation being the active conformation as opposed to the same-gyres conformation.

The gaps data are presented as evidence for ATPase binding to the SHL2 location on the same gyre as the entry DNA.

ii) Previous work has shown that Chd1 can move nucleosomes by 23-29 base-pairs without a stall, which is more than the 7 bp that is contacted by the DBD. The authors use this data as evidence against the cross-gyres conformation. This reasoning is not clear because the ISWI ATPases (Deindl et al., 2013) have step sizes as assessed by smFRET that are not correlated with the amount of DNA that is contacted by their DBDs. Further depending on the processivity of the enzyme the amount of DNA moved before a stall could be several multiples of 7 bp.

This is a good point, the difference in the initial and subsequent step sizes observed in the Deindl paper is mentioned in the revised Discussion.

iii) In the Discussion, the authors say: "This is anticipated if the DNA binding domain acts to generate an active conformation, but not if sensing of exit linker DNA is repressive." However, the ATPase data from Nodelman et al. (2017) shows a repressive effect of exit linker DNA. This should be discussed.

This data is mentioned in the revised Discussion.

iv) In the Discussion, the authors say: "Thirdly, the activity of chimeric Chd1 proteins in which DNA binding is provided via a heterologous domain is greatest when the cognate binding site is placed in the entry linker mimicking the arrangement observed in the Chd1-nuceosome structure". While this data is consistent with the cross-gyres conformation being the active conformation, the data is also consistent with the same-gyres conformation being active if what is being trapped by EM is the inactive state. In other words this prior data does not rule out the same-gyres conformation being the active conformation.

This is a good point we removed this section.

6) A few straightforward biochemical tests are required to test the mechanistic significance of the structural data. The authors can investigate the functional effects on nucleosome sliding of mutating the Chd1 lobe II residues that make contacts with the H3-alpha1 helix and the 25 N-terminal residues that are hypothesized to contact the unpeeled DNA.

New data has been added showing that deleting the lobe II residues abolishes Chd1 activity in nucleosome sliding (Figure 2—figure supplement 5). The truncation of the H3 tail to K36 unexpectedly stimulates Chd1 activity quite strongly. However, deletion to 26 has no effect. This indicates that the interaction of the N-terminal region of the tail with DNA is not required for it to exert a repressive effect. This shows an important role for the histone H3 N-terminal tail in Chd1 action. The new data is included as Figure 8—figure supplement 1.